# Effect of Whey Protein Concentrate on Physicochemical, Sensory and Antioxidative Properties of High-Protein Fat-Free Dairy Desserts

**Katarzyna Kusio** [1,2]**, Jagoda O. Szafrańska** [2] 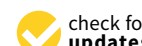**, Wojciech Radzki** [3] **and Bartosz G. Sołowiej** [2,]*

1    Hotel and Catering School, Św. Brata Alberta 1, 26-600 Radom, Poland; k.kusio@wp.pl
2    Department of Milk Technology and Hydrocolloids, Faculty of Food Sciences and Biotechnology, University of Life Sciences in Lublin, Skromna 8, 20-704 Lublin, Poland; jagoda.szafranska@poczta.fm
3    Department of Fruits, Vegetables and Mushrooms Technology, Faculty of Food Sciences and Biotechnology, University of Life Sciences in Lublin, Skromna 8, 20-704 Lublin, Poland; wojciech.radzki@up.lublin.pl
*    Correspondence: bartosz.solowiej@up.lublin.pl; Tel.: +48-81-4623350

**Abstract:** This study evaluates a new formula for high-protein fat-free dairy desserts. The rheological, textural, organoleptic and antioxidative properties of this product have been examined. They were prepared in laboratory conditions using a magnetic stirrer and then mixed in a water bath at 80 °C for 10 min. The composition included skimmed-milk powder, different concentrations of whey proteins (WPC80) (5, 7, 9, 11 or 13%), sucrose and k-carrageenan. Samples were stored at 4 °C. The use of different amounts of whey proteins significantly influenced the texture, rheological properties and appearance of dairy desserts. With the increase of WPC80 content, the hardness (5–11%), adhesiveness (5–13%), cohesiveness (513%), springiness (5–11%) and viscosity (5–13%) of the high-protein fat-free dairy desserts increased. Elastic (gel) properties were exhibited throughout the process (G′ > G″). This result was confirmed by the measurements of tan(δ) (tan(δ) < 1). Water activity decreased with an increase in WPC80 content. Health-promoting features using 2,2-diphenyl-1-picrylhydrazyl (DPPH) and ferric reducing antioxidant power (FRAP) measurements were evaluated. Both methods showed that the samples have antioxidant properties. Panelists described desserts with 9% WPC80 as the most favorable according to sensory properties. This research promotes the reduction of production waste using high-protein whey powder, a by-product of whey from cheese manufacturing, as a main component during food production, and it also promotes fat-free food.

**Keywords:** dairy dessert; whey proteins; fat-free; high-protein; texture; viscoelastic properties; water activity; sensory; antioxidative properties

## 1. Introduction

The dairy desserts market is expected to grow in the next few years thanks to new nutritional trends searching for ready-to-eat food containing health-promoting components that are also low in fat. This type of product is abundant in calcium, potassium and vitamins, which fits well with the demand for health-promoting, low-fat desserts. Growing interest in new dairy products forces producers to invent new formulas and innovations [1].

Dairy desserts have a gel or semi-liquid texture [2]. They are formulated from whole and/or skimmed milk, sucrose, thickeners such as starch and hydrocolloids, aroma and colorants [3]. Due to their nutritional value, they are widely consumed by different groups of people. Producers prepare appropriately modified products for children, adults and elderly people [3–5]. The rheological and textural properties are closely related to the ingredients used to prepare the dairy desserts.

These features are essential in terms of industrial manufacturing, nutritional characteristics and consumer acceptance [3,6].

Casein has been the most extensively studied protein in recent years. This interest relates to its huge potential for applications as an addition to different products in the food industry. It is one of the major proteins in milk and has excellent functionalities, such as emulsification and gelling capabilities [7]. Additionally, casein can inhibit fecal beta-glucuronidase, an enzyme that deconjugates procarcinogenic glucuronides to carcinogens. It stimulates phagocyte activities and increases the ability of lymphocytes to protect organisms against colon cancer [8]. Whey proteins are another major protein fraction in milk. It is a product primarily obtained by the dairy industry during the production of cheese and casein. It consists mainly of water (94%), lactose, proteins and fats. 70–80% of all proteins found in whey mass are α-lactalbumin and β-lactoglobulin. Other components include bovine serum albumin (BSA) and bovine lactoferrin (BLF), immunoglobulins (IG) and lactoperoxidase (LP) [9]. Caseins and whey proteins are characterized by specific biological and physiological properties. In recent years, researchers have investigated the therapeutic aspects of milk proteins. For example, it has been proven that biologically active peptides isolated from whey lower blood pressure and reduce total cholesterol, LDL and triglycerides, which are connected with the risk factors of cardiovascular disease [10]. Whey-protein concentrate in moderate concentrations promotes glutathione production, which in turn enhances the antioxidant activity in a pheochromocytoma (PC12) cell line [9]. It has also been confirmed that milk proteins, especially whey proteins, may have an influence on the protection of the human body against different types of cancer, such as breast, colon or prostate gland cancer. This ability is associated with the increase of cellular glutathione levels and the promotion of hormonal and cell-mediated immune reactions [8,11]. Davoodi et al. [8] reported that several in vitro and in vivo studies had found evidence that whey proteins could positively impact human immune responses. They observed the inhibitory activity of whey proteins against *Helicobacter pylori* in infected subjects. Furthermore, Okuda et al. [12] found that the oral administration of lactoferrin tablets taken twice a day for twelve weeks reduced the capacity of *H. pylori* to form colonies. Moreover, milk proteins are important additives in dairy-industry production due to properties such as solubility, high nutritional value, their bland flavor profile and their ability to reduce fat content [13].

Reis et al. [6] suggested that the consumption of products containing saturated fat is connected with heart disease. Therefore, there is a need for new food systems either containing lower amounts of fat or that are fat free [6]. From a technological point of view, fat is an important ingredient that affects the taste of a product, but also improves the texture and mouthfeel. Products without this component are tasteless and unacceptable to consumers. Taking this into account, during the preparation of our product we used WPC80, which is considered to be fat mimetic [14]. It has various functional properties similar to those of lipids. Whey proteins can completely or partially replace hydrocolloids or modified starch and provide health-promoting effects. The most important technological functions of WPC in products with reduced fat content are: water binding, emulsification, gelation and increased viscosity [14]. Kappa carrageenan or its kappa–iota hybrids is a specific food additive because of its properties that differentiate it from those of other hydrocolloids, for example, its ability to form gels with calcium and potassium as well as its essential feature of reactivity with milk proteins. The kappa form of carrageenan gels in the presence of either $K^+$ and/or $Ca^{2+}$ ions, while iota gels only in the presence of $Ca^{2+}$ [15]. For all dairy products, including desserts, the structure of the finished product determines the quality. Solubility, flow, viscoelastic and fracture parameters are important features, especially in modified products with different additives [13]. Dairy desserts are a type of product that show shear-thinning flow behavior and viscoelastic properties typical of weak gels. However, noticeable differences in rheological properties can be found in different model systems with new compositions [4,5,16]. When new additives and ingredients, or their combinations, are incorporated into products, the effect of such modification on the textural and rheological properties of the final product should be researched. The authors attempted to obtain a fat-free dairy dessert through the addition of low-protein whey powder in low concentrations (1.5–4.5%) and evaluated the textural

and organoleptic characteristics of the above-mentioned product [17]. To the best of our knowledge, there are no studies concerning the antioxidative properties, oscillatory rheometry and water activity of fat-free dairy desserts containing high-protein whey powders. These parameters are important in food product development (rheological properties: G′, G″, etc.) for the enhancement of the health-promoting properties of food products (antioxidative properties), food processing and storage (water activity). Therefore, the purpose of this study was to prepare and carry out experiments and to analyze findings related to the physicochemical, sensory and antioxidative properties of high-protein fat-free dairy desserts with a whey-protein concentrate (WPC80) as a fat replacer.

## 2. Materials and Methods

### 2.1. Materials

The following raw materials were used for the production of high-protein fat-free dairy desserts: skimmed milk powder (SMP, 34.2% protein) (Polsero, Sokołów Podlaski, Poland), whey-protein concentrate (WPC80, 76.8% protein) (Milei, Leutkirch, Germany), sucrose (Krajowa Spółka Cukrowa, Krasnystaw, Poland) and k-carrageenan (Tic Gums, Belcamp, Maryland, MA, USA).

### 2.2. Preparation of High-Protein Fat-Free Dairy Desserts

WPC80 (5% w/w) was dissolved in distilled water using a magnetic stirrer (Heidolph MR 3002S, Schwabach, Germany) (temp. 21 °C, 300 rpm). Other samples were prepared with 7%, 9%, 11% or 13% WPC80. Then, 3.33% of sucrose, 3.33% of skimmed-milk powder (SMP) and 0.05% of k-carrageenan were placed in a container and mixed with a magnetic stirrer (Heidolph MR 3002S, Schwabach, Germany) at room temperature. Kappa-carrageenan was dissolved in 90 mL of water, which had been prepared separately beforehand using a magnetic stirrer (Heidolph MR 3002S, Schwabach, Germany) at room temperature and then heated for 15 min at approx. 70 °C while stirring. Then, the pH of the WPC80, SMP, sucrose and k-carrageenan mixture was adjusted to 6.2 using citric acid or sodium hydroxide (2 mol/L). After that, the mixture was immersed in an 80 °C water bath (to improve the mixing process and ensure pasteurization) and the contents were then mixed at 10,000 rpm for 10 min with the H 500 homogenizer (Pol-Eko Aparatura, Wodzisław Śląski, Poland). Ready desserts were poured into plastic containers (cylindrical, sample size—40 mm in diameter and 40 mm in height). Each sample was prepared in triplicate. The samples were stored at room temperature for 30 min and then overnight at 4 °C. The dairy desserts were taken from the refrigerator 1 h before measurement to reach a temperature of 21 °C. The composition of high-protein fat-free dairy desserts is presented in Table 1.

**Table 1.** Composition of high-protein fat-free dairy desserts.

| Ingredients | Quantity (%) | | | | |
|:---:|:---:|:---:|:---:|:---:|:---:|
| WPC80 | 5.0 | 7.0 | 9.0 | 11.0 | 13.0 |
| SMP | | | 3.33 | | |
| sucrose | | | 3.33 | | |
| k-carrageenan | | | 0.05 | | |
| Water | 88.29 | 86.29 | 84.29 | 82.29 | 80.29 |

### 2.3. Penetration Test

All measurements were carried out using the TA-XT2i Texture Analyzer (Stable Micro Systems, Godalming, Surrey, UK). The high-protein fat-free dairy desserts were penetrated by a 15 mm diameter cylindrical probe to the depth of 28 mm, as specified by Szafrańska, Muszyński, and Sołowiej [18]. The penetration rate was 1 mm/s. The dairy desserts were evaluated for their hardness, adhesiveness,

cohesiveness and springiness using Texture Expert software (v. 1.3, Stable Micro Systems, Godalming, Surrey, UK). Five measurements were carried out for each of the three replicates.

## 2.4. Viscosity Measurements

The apparent viscosity of high-protein fat-free dairy desserts was investigated using a Brookfield DV II+ rotational rheometer (Brookfield Engineering Laboratories, Stoughton, MA, USA) equipped with a Helipath Stand and T-bar spindle D. Measurements were carried out at 21 °C with a spindle velocity of 0.5 rpm.

## 2.5. Viscoelastic Properties

The storage (G′) and loss (G″) moduli and the tan(δ) of the desserts were measured applying a Kinexus lab+ rheometer (Malvern Panalytical, Cambridge, UK) using serrated plates (PU40X SW1382 SS and PLS40X S2222 SS at plate–plate configuration). Measurements were made at 21 °C and a frequency of 0.1 Hz. The results from the measurements were computer-registered in the Kinexus Malvern program rSpace (Malvern Panalytical, Cambridge, UK).

## 2.6. Back Extrusion

The desserts were tested with a TA-XT2i Texture Analyzer (Stable Micro System, Godalming, Surrey, UK) by back extrusion. Stable Micro system tooling (back extrusion ring) was used with a container diameter of 50 mm and a head diameter of 45 mm. Head travel speed was 1 mm/s. The research was carried out in three repetitions.

## 2.7. Water Activity

The water activity ($a_w$) of the desserts was determined using the AWMD-10 water activity meter (NAGY Messsysteme GmbH, Gäufelden, Germany) with an accuracy of ±0.001 of an $a_w$ unit. Before measurement, the apparatus was calibrated to the dedicated humidity standard (95% HR). Measurements were taken at 25 °C in five repetitions.

## 2.8. Solvent Extraction for Antioxidant Assays

1 g of each dessert was suspended in water (15 mL) and disintegrated with a T10 dispersing tool (Ika, Staufen, Germany). This was followed by water extraction carried out at 50 °C for 60 min using a shaker (Incu-Shaker Mini, Benchmark Scientific, NJ, USA) set at 150 rpm. The obtained slurries were centrifuged at 21,000× $g$ for 10 min at 4 °C and the extracts were subjected to antioxidant assays.

## 2.9. Determination of Antioxidant Properties by the DPPH Method

The antioxidant properties of the desserts were determined using 2,2-diphenyl-1-picrylhydrazyl (DPPH) analysis as described by Blois [19]. The extracts (0.2 mL) were mixed with 0.8 mL of 0.2 mM DPPH (Sigma-Aldrich, St. Louis, MO, USA) ethanolic solution. The mixtures were vortexed and incubated for 15 min in the dark. The absorbance was then measured at 520 nm against a blank sample with a UV–vis spectrometer (Helios Gamma, Thermo Fisher Scientific, Waltham, MA, USA). The antioxidant capacity of the extracts was calculated from the obtained calibration curve prepared with known concentrations of Trolox. The results are presented in micromoles of Trolox per gram of dessert.

## 2.10. Determination of Antioxidant Properties by the FRAP Method

The ability of these desserts to reduce ferric ions was analyzed according to the spectrophotometric method described by Benzie and Strain [20]. Fresh ferric reducing antioxidant power (FRAP) reagent was prepared by mixing a 300 mM acetate buffer at pH 3.6 with a 2,4,6-tri(2-pyridyl)-1,3,5-triazine (TPTZ) (Sigma-Aldrich, St. Louis, MO, USA) solution (10 mM TPTZ in 40 mM HCl) and a 20 mM $FeCl_3 \cdot 6H_2O$ (POCH, Gliwice, Poland) solution at a 10:1:1 ratio. For this purpose, 0.1 mL of each sample

were mixed with 1.9 mL of the FRAP reagent solution. The mixtures were vortexed and incubated for 15 min at 37 °C. The absorbance was measured at 593 nm using a UV–vis spectrophotometer (Helios Gamma, Thermo Fisher Scientific, Waltham, MA, USA). The results were read from the Trolox standard curve and the data reported as micromoles of Trolox per gram of dessert.

### 2.11. Organoleptic Evaluation

The sensory features of the fresh samples of the desserts were evaluated by 10 untrained panelists, all staff members of the Faculty of Food Sciences and Biotechnology at the University of Life Sciences in Lublin, Poland. The organoleptic evaluation of these high-protein fat-free dairy desserts included an evaluation of appearance, color, texture and flavor. The presence of air bubbles, syneresis and protein clots were also considered. The evaluated features of the prepared product were as follows: flavor (1–10 points) and body, texture, appearance and color (1–5 points), in accordance with Szafrańska and Sołowiej [21].

Ethics

Only adults participated in the sensory study of high-protein fat-free dairy desserts. The Local Bioethics Committee in Lublin, Poland, concluded that the above study did not require the consent of the Commission. This study did not have any predictable risks, nor did it expose participants to pain. No personal or identifying information was collected and all data were analyzed anonymously. As such, while written participant consent was not collected for this study, all participants gave verbal informed consent to participate. They were informed of the nature of the study and its objectives and advised of participants' confidentiality and anonymity. To make participants feel comfortable, they were allowed to withdraw from the study at any time and for any reason. All participants evaluated the tested products objectively and agreed to the publication of their evaluation results, which would remain anonymous.

### 2.12. Statistical Analysis

The statistical analysis was carried out with the help of the STATISTICA 12.0 PL software (Stat Soft Polska Sp. z o. o., Kraków, Poland). A one-way ANOVA analysis was performed and significant differences between samples were determined by the Tukey post hoc test at $p < 0.05$.

## 3. Results and Discussion

### 3.1. Penetration Test

Whey proteins, alone or in combination with other carbohydrate-based ingredients, give food a creamy texture and affect particular rheological features of the products [22]. Table 2 presents the effect of different WPC80 quantities (5–13%) on the texture attributes (hardness, adhesiveness, cohesiveness and springiness) of high-protein fat-free dairy desserts. We have taken into consideration that carrying out a penetration test inside a plastic container has an additional impact on the measurements because of the effect of the container walls. However, the penetration test or modified texture profile analysis (TPA) applies to different semi-solid dairy products, e.g., desserts, processed cheese, cheese sauces, etc.

With the increase in WPC80 content, the hardness (5–11%), adhesiveness (513%), cohesiveness (513%) and springiness (5–11%) of these high-protein fat-free dairy desserts also increased. The highest values for the adhesion (542.9 J) and cohesion (0.52) of the product were observed in dairy desserts with 13% WPC80 ($p < 0.05$). The lowest values of the examined features related to products with the lowest (5%) addition of WPC80 (Table 2).

**Table 2.** Effect of different WPC80 contents on the texture attributes of high-protein fat-free dairy desserts.

| Content of WPC80 [%] | Texture Attributes | | | |
| --- | --- | --- | --- | --- |
| | Hardness [N] | Adhesiveness [J] | Cohesiveness | Springiness |
| 5 | 21.56 [a] ± 0.78 | 58.25 [a] ± 4.22 | 0.47 [a] ± 0.02 | 0.94 [a] ± 0.009 |
| 7 | 35.78 [b] ± 1.68 | 169.59 [b] ± 10.74 | 0.48 [ab] ± 0.03 | 0.94 [a] ± 0.004 |
| 9 | 37.45 [b] ± 1.46 | 197.09 [b] ± 17.03 | 0.51 [ab] ± 0.02 | 0.95 [a] ± 0.005 |
| 11 | 94.2 [d] ± 5.54 | 412.36 [c] ± 28.75 | 0.51 [ab] ± 0.02 | 0.98 [c] ± 0.02 |
| 13 | 71.53 [c] ± 4.41 | 542.9 [d] ± 29.5 | 0.52 [b] ± 0.03 | 0.97 [b] ± 0.03 |

Data is presented as means ± SD (standard deviation). [a–d] Means in the same column with different superscripts are significantly different ($p < 0.05$, Tukey's HSD test).

Hardness is a specific force required to deform food between the teeth or between the tongue and palate [23]. Hardness was the dominant feature among other texture parameters and was positively associated with them. A uniform increase in product hardness was observed in the range between 5 and 11% of the added product (21.56–94.2 N). Gustaw and Nastaj [24] examined changes in yogurt hardness caused by the addition of selected milk preparations. They observed that the highest values of hardness were found in the product with added WPC85 among other tested ingredients (OMP, WPC35, WPC65, degreased WPC (WPCO)). They also suggested that the increase in hardness of tested yogurts can be explained by the correlation between the content of casein and whey proteins. The structure of the final products became more compact and therefore harder [24]. The results obtained in our research confirm this conclusion, but a decrease in hardness was noted in the product with 13% WPC80. This might be caused by the type of whey concentrate used in our experiment. WPC80, among other whey-protein concentrates (WPC35, WPC50 and WPC65), has the lowest moisture and lactose content and the highest protein content [25]. This composition together with higher amounts of added WPC80 caused a substantial increase in the product hardness with 5, 7, 9 and 11%, but had no influence on the product with 13% WPC80. The decrease in hardness of the dairy dessert with 13% WPC80 can be ascribed to phase separation between whey proteins and k-carrageenan. Thermodynamic incompatibility (phase separation) is the most common phenomenon occurring between proteins and hydrocolloids in a system where pH surpasses the protein isoelectric point [26,27], which is in line with our study, where the pH was 6.2. A greater amount of added WPC80 increased the surface-active material in the system and the number of protein interfacial shells [28]. Szafrańska et al. [18] tested the effect of WPC80 on processed cheese sauces. They also noticed that in a product with added coconut oil, not only were the values of hardness lower in samples with the highest amount of added whey-protein concentrate compared to a product with a lower amount of WPC80, but also the adhesiveness of the samples. Adhesiveness is a force that is required to remove food from the palate or teeth while chewing [23]. The quantity and hydrophobicity of proteins are not always the factors that determine the highest adsorption at fat–water interfaces during emulsification [18]. Moreover, whey proteins perform in the same way as fat globules, thus, the production of high-protein desserts without fat can be achieved. This, however, increases product adhesiveness [22].

As other studies show, the incorporation of whey-protein concentrate increases not only the hardness in yogurts [29], but also the adhesiveness [30]. Herrero and Requena [30] investigated the effect of supplementing the set-type yogurt made from goat's milk with whey protein concentrate. They reported that WPC35 enhanced the firmness, hardness and adhesiveness during a 28-day storage period.

Cohesiveness is a representation of the forces of internal bonds holding the product as a whole together [31,32]. The lowest cohesion was measured in a product with a 5% protein addition of WPC80, while the values of products containing 7, 9 and 11% concentrations of whey remained at the same level ($p > 0.05$). Only the highest level of WPC80 (13%) caused a significant increase in the cohesiveness of the tested product. These results may suggest that increasing the protein concentration in a product has little effect on the increase in cohesion values. In research into the storage stability of the texture

and sensory properties of yogurt with the addition of polymerized (PWP) and non-polymerized (WPC80) whey proteins, the authors concluded that if they had used any of the tested forms to produce the final product, the consistency of the yogurts would have been significantly greater than that of the control sample. The addition of WPC80 maintained the original cohesiveness of the yogurt for longer than the control yogurt during storage [33]. These results are similar to ours, but we did not note such a great increase in the tested feature. This could have been caused, similarly to the product hardness tests, by the WPC80 we used in our research and its composition.

The obtained results could also be influenced by the reactions taking place between whey proteins and casein micelles induced by the heat. Researchers noticed that a reaction that takes place in milk and in re-suspended casein micelles during the heating process up to 90 °C causes limitations in the interactions between β-lactoglobulin and micelles because of the low number of binding sites available. Additionally, the amount of α-lactoglobulin associated with the micellar pellet seemed to be related to its concentration in milk [34].

Springiness is the speed at which a deformed product returns to its initial state [31]. In texture profile analysis (TPA), springiness (elasticity) is defined as the total deformation of the specimen in the second bite [35]. In our experiment, a noticeable ($p < 0.05$) increase in this parameter was observed in samples with the addition of 11% and 13% WPC80 (0.98 and 0.97, respectively) ($p < 0.05$). In 2018, researchers examined the whey protein–pectin gel structure and determined the effect of heat treatment on WPC80 (50–90 °C). They concluded that an increase in heat treatment temperature on whey proteins resulted in an increase of springiness value in pectin–whey protein gels. They also proved that the highest value of this feature was obtained at temperatures between 80 and 90 °C. This may suggest that the amount of added protein had a smaller effect on the tested property compared to the temperature used during mixing [36].

### 3.2. Viscosity Measurements

Figure 1 shows the results of measuring the apparent viscosity of the high-protein fat-free dairy desserts. Increases in protein concentration increased the viscosity of the tested products. The lowest values of the tested samples were observed in the product with the addition of 5 and 7% WPC80 (100 J and 130.7 J, respectively) and the highest in the product with 13% WPC80 (507.3 J) ($p < 0.05$). The obtained results and visual observations of our tested products allowed us to conclude that the highest and the lowest values of the tested feature (viscosity) did not resemble the texture of any of the commercial dairy desserts that we wanted our product to be similar to.

Viscosity of liquid food can be described as a resistance to flow, informally known as 'thickness' [37]. This feature in food products could be affected by many factors such as temperature, hydrocolloid concentration, molecule size and complexity and polymer chain length in foods such as proteins, starches, hydrocolloids or gums [38]. There are not many studies describing the results of research on the effects of whey proteins as a total replacement for fat in the preparation of dairy desserts. Most research focuses on the partial replacement of fats with the addition of described proteins. One example is a yogurt obtained by the thermostat method with the addition of 85% whey-protein concentrate compared to the addition of 35 and 65% WPC, which was characterized by the highest viscosity and an appropriate texture [24]. The longer heating of milk desserts at a temperature of 85 °C resulted in an increase in viscosity compared to heating in the range of 60–85 °C. This is due to the gelling properties of whey proteins. During the cooling of WPC desserts from 85 °C to 27 °C, a slow increase in viscosity was observed, but the highest increase in this parameter was observed in temperatures of 24–26 °C [2]. Moreover, the interactions of α-lactalbumin, β-lactoglobulin and casein during the heat treatment can increase the viscosity of tested products [33].

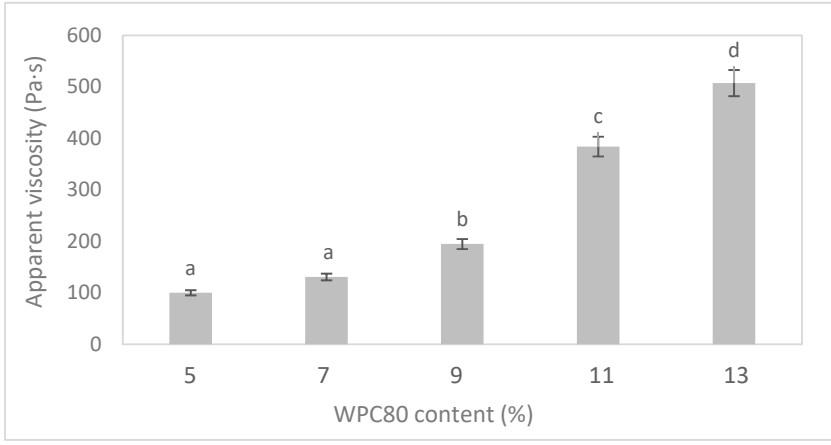

**Figure 1.** Effect of different WPC80 contents on the apparent viscosity of high-protein fat-free dairy desserts. Letters (a–d) indicate significant differences at $p < 0.05$ (Tukey's HSD test).

*3.3. Viscoelastic Properties*

Because the high-protein fat-free dairy dessert structure can be described as gel-like, we can measure G′ and G″ parameters. The storage modulus (G′) represents energy that is retained by the system as a result of elastic deformation during the temporal application of stress, and the loss modulus (G″) reflects the ratio of the viscous stress amplitude to the strain amplitude and indicates how much energy during deformation has been dissipated as heat [38]. The ratio of loss to storage modulus is represented by tan(δ), which provides information about the number of elastic and viscous components contained in the sample [39]. Increased G′ values along with the addition of WPC80 (5–11%) testify to the strengthening of the dessert gel structure ($p < 0.05$). Only in a product with 13% WPC80 was a decrease in the value of the measured feature recorded (Table 3). In all tested samples, the values of the storage (G′) modulus were always higher than those of the loss (G″) modulus. It means that the high-protein fat-free dairy desserts exhibited elastic (gel) properties throughout the measurement. Samples with 13% WPC80 had lower G′ and G″ values than the product with 11% WPC80, indicating that the gel structure of the tested product had weakened and formed a less elastic system.

**Table 3.** Effect of different WPC80 contents on storage (G′) and loss (G″) moduli, and tan(δ) of high-protein fat-free dairy desserts.

| Content of WPC80 [%] | G′(Pa) ± SD | G″(Pa) ± SD | tan(δ) ± SD |
|:---:|:---:|:---:|:---:|
| 5 | 262.81 [a] ± 3.88 | 73.12 [a] ± 0.05 | 0.28 [a] ± 0.05 |
| 7 | 502.2 [b] ± 6.77 | 148.95 [b] ± 22.77 | 0.3 [a] ± 0.046 |
| 9 | 867.07 [c] ± 40.73 | 342.99 [c] ± 8.94 | 0.4 [b] ± 0.02 |
| 11 | 1608.6 [e] ± 75.33 | 488.15 [d] ± 67.56 | 0.31 [a] ± 0.06 |
| 13 | 1119.15 [d] ± 76.99 | 557.86 [e] ± 29.93 | 0.5 [c] ± 0.06 |

Data is presented as means ±SD (standard deviation). [a–e] Means in the same column with different superscripts are significantly different ($p < 0.05$, Tukey's HSD test).

A correlation of storage (G′) and loss (G″) moduli with hardness values indicates that the structure of the obtained product has become harder and more compact with the addition of WPC80 (5–11%). A sample of the dairy dessert with 13% WPC80 presented lower values than the product with 11% WPC80. When the loss value is tan(δ) < 1 the tested dairy dessert exhibits elastic (gel) properties; otherwise (i.e., tan(δ) > 1) the product shows viscous properties. In every tested product, the value of tan(δ) was lower than 1. The highest values of loss modulus were noticed for the 13% WPC80 concentration. Our results are consistent with the conclusions of other researchers who confirm the relationship between the amount of protein and values of storage (G′) and loss (G″) moduli [40].

This is because, with a low lactose content in concentrates and isolates, when the whey proteins are denatured, their gelation temperature is shifted [41]. We also noticed a significant difference in the values between 13% and 11% WPC80 ($p < 0.05$). The decrease in this value may be related to the foamability of the whey proteins. As for measurements describing a product's hardness, a greater amount of added whey-protein concentrate increased the surface-active material in the food system and produced more protein interfacial shells [28].

### 3.4. Back Extrusion

The technique of back extrusion is often used to test foods to identify the flow behavior of complex fluids. This method estimates the rheological properties of products under sticking boundary conditions [42].

Figure 2 presents the results of the measuring force values of the desserts using back extrusion. In the tests carried out using the back-extrusion method, it was shown that the increase in the force values of the tested products correlated with the increase in the content of the WPC80 addition in the 5–11% range. The sample with 13% WPC80 had a lower value (436 g) compared to the dairy dessert with 11% (499 g). Therefore, these results indicate the formation of an increasingly dense gel structure with the increasing content of whey-protein concentrate. They correlate with storage (G') and loss (G") moduli results as well as hardness. There is little research discussing the reasons for an uneven increase in product hardness relating to supplied whey proteins. Carrageenan can interact with proteins through electrostatic or ionic bonding. Its molecules carry negative charges and can therefore be combined with positively charged proteins or particles like potassium (c.a. 520 mg), which is found in WPC80. For example, when carrageenan combines with a milk protein such as casein it can form a three-dimensional gel network [43]. Additionally, Gustaw et al. [2] suggested that carrageenan affects the increase in the hardness of dairy desserts, but at a higher concentration it may cause adverse changes in the tested sample texture due to the interaction between milk proteins and carrageenan [2]. Other studies have suggested that the rheological properties of gels prepared with the addition of carrageenan may be influenced by the separation of the phase between carrageenan and the proteins. This can occur between β-lactoglobulins and k-carrageenan based on depletion interactions in a suspension of small proteins immersed in a solution of polysaccharides that then form an entangled network [26]. These theories can explain the decrease in the back-extrusion measurement value while the hardness of the product increases with the addition of 13% WPC.

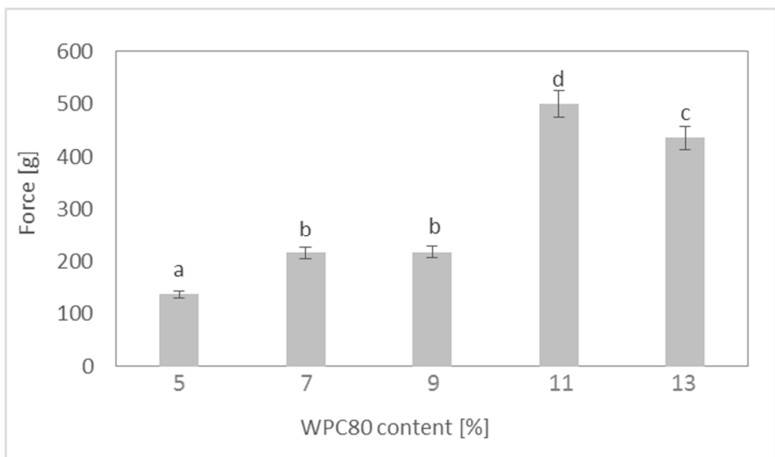

**Figure 2.** Effect of different WPC80 contents on force values of high-protein fat-free dairy desserts. Letters (a–d) indicate significant differences at $p < 0.05$ (Tukey's HSD test).

### 3.5. Water Activity

Water activity ($a_w$) is the ratio of vapor pressure and vapor saturation pressure at the same temperature. This measurement is used to control the quality and stability of foods in terms of microorganism growth, enzymatic reaction rates and physical properties [44]. The results of our measurements are presented in Figure 3. The water activity of the high-protein fat-free dairy desserts with a different WPC80 content decreased significantly ($p < 0.05$) with higher WPC80 concentrations. The highest values of $a_w$ were observed in the sample with 5% WPC80 (0.975) and the lowest in the product with 13% WPC80 (0.938).

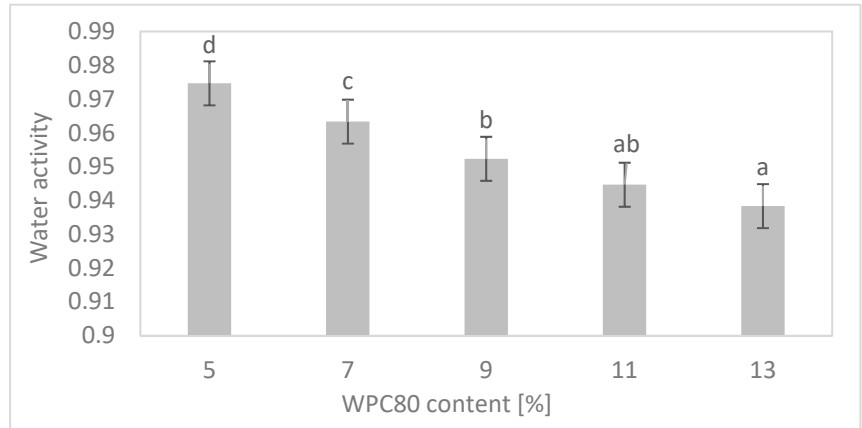

**Figure 3.** Effect of different WPC80 contents on water activity of high-protein fat-free dairy desserts. Letters (a–d) indicate significant differences at $p < 0.05$ (Tukey's HSD test).

A typical shelf-stable dairy product should have low water activity (i.e., $a_w$ less than 0.85). Often, products at this level of $a_w$ are characterized by an undesirable texture that can be eliminated by using different specific functional and/or processing parameters. For example, aerated dairy products generally have a water activity value ($a_w$) above 85%, which is associated with the need for refrigerated storage. Even when producers and consumers fulfill this requirement, the product often has a relatively short storage life [45]. A significant decrease ($p < 0.05$) in the value of water activity in the tested products along with an increase in the content of whey-protein concentrate may be caused by a higher amount of proteins in the formula and the functional features of WPC80, such as its water binding properties [46].

### 3.6. Determination of Antioxidant Properties by DPPH and FRAP Methods

Antioxidant activity is a complex procedure and is influenced by many factors that cannot be described with only one single method. Therefore, it is important to perform more than one type of antioxidant capacity measurement to take into consideration various mechanisms of antioxidant action. In the study of these high-protein fat-free dairy desserts, antioxidant activity was tested using different methods (DPPH and FRAP). Applying the DPPH method, it was possible to measure the ability of the antioxidants to donate an electron or hydrogen radical to the stable DPPH free radical. Meanwhile, the FRAP method made a comparison of antioxidants based on their ability to reduce ferric ($Fe^{3+}$) to ferrous ($Fe^{2+}$) ions through the donation of an electron [47]. The results of our experiments are presented in Figure 4 (DPPH measurements) and Figure 5 (FRAP measurements).

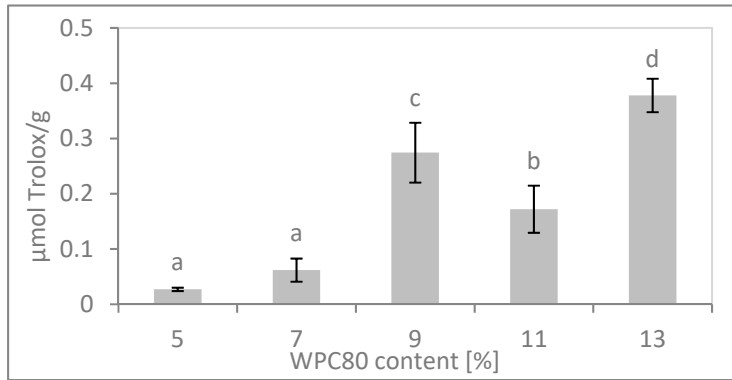

**Figure 4.** Effect of different WPC80 contents on antioxidant properties (DPPH) of high-protein fat-free dairy desserts. Letters (a–d) indicate significant differences at $p < 0.05$ (Tukey's HSD test).

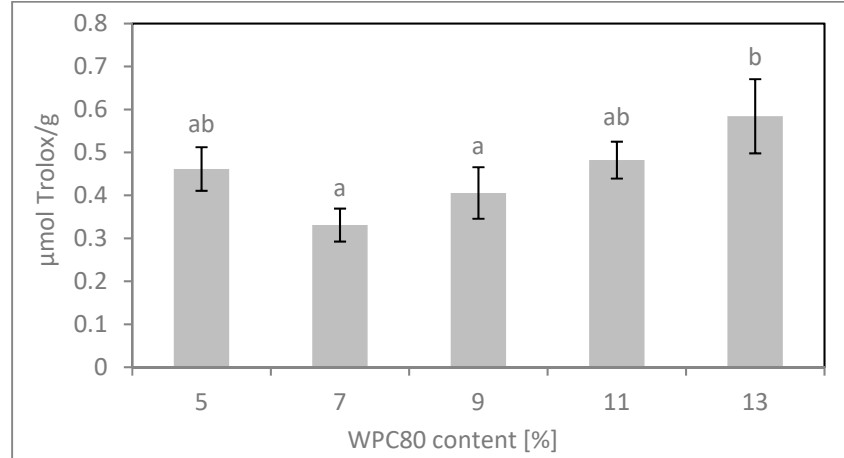

**Figure 5.** Effect of different WPC80 contents on ferric reducing antioxidant power (FRAP) of high-protein fat-free dairy desserts. Letters (a–b) indicate significant differences at $p < 0.05$ (Tukey's HSD test).

Both selected spectrophotometric methods for determining the antioxidant properties follow the same reaction mechanism, i.e., electron transfer from antioxidant to oxidant ($Fe^{3+}$ and DPPH radical). All tested samples could reduce the stable, purple-colored radical DPPH into the yellow-colored DPPH-H. An increase in the value of measurement (µmol Trolox/g) along with the increase in value of WPC80 content (5–9%) was observed (Figure 4). The highest value of the measured parameter was observed in the sample with 13% WPC80 (0.38 µmol Trolox/g) and the lowest values were obtained in the product with 5% WPC80 (0.027 µmol Trolox/g). Unal and Akalm [48] tested the antioxidant activity of yogurt fortified with sodium and calcium caseinate and 2 or 4% whey-protein concentrate. Their results suggested that the product fortified with 4% WPC80 had the highest ($p < 0.05$) scavenging activity. They did not report significant differences among yogurt samples with the addition of lower amounts of this ingredient ($p > 0.05$). They concluded that the scavenging effect of whey proteins could be attributed to lactoferrin, which has been reported as an important component for high scavenging activity [48].

The FRAP method examined the ability of the sample to reduce the ferric ion, which is a criterion of the antioxidative capacity [49]. This method has already been used to characterize the antioxidative capacity of a dairy product, e.g., whey proteins [50]. An analysis of the antioxidant properties of the high-protein fat-free dairy desserts was presented in Figure 5. In general, the values described in individual products remained at a similar level. However, the lowest values were obtained in the product with 7% WPC80 (0.33 µmol Trolox/g) and the highest in the sample with 13% WPC80 (0.58 µmol Trolox/g).

Literature describes lactoferrin as a whey compound with a strong antioxidative capacity [50–52]. Different studies suggest that the addition of whey-protein concentrate to dairy products significantly increases the antioxidative properties of the tested samples [51]. It is worth mentioning that the FRAP and DPPH methods are in vitro assays conducted without alternative targets and obtained data cannot be readily translated to more complex systems. Despite this, researchers suggest that whey proteins have the potential to survive in the gastrointestinal tract and the bloodstream post-consumption [52].

In the present research, the data obtained from the DPPH and FRAP methods correlated well (correlation coefficient, R = 0.84), which is in agreement with many previous studies [53–55]. Some discrepancies between the data from the FRAP and DPPH antioxidant methods occurred. However, such discrepancies are not uncommon and have been reported previously with different biological material [55–57]. Moreover, there are reports in which there is no correlation, or the correlation is weak [58]. Although both the FRAP and DPPH assays are based on the transferring of electrons from the antioxidant and reduction of oxidant, the FRAP method is more prone to color interference and slow color development. It also involves more antioxidant compounds that react with various kinetics [59,60].

### 3.7. Organoleptic Evaluation

The sensory characteristics of these high-protein fat-free dairy desserts result from the composition of our product formula. The source of protein and lack of typical fat source had a great impact on the taste and body features of the tested samples. The sensory scores for each of the desserts are shown in Figure 6.

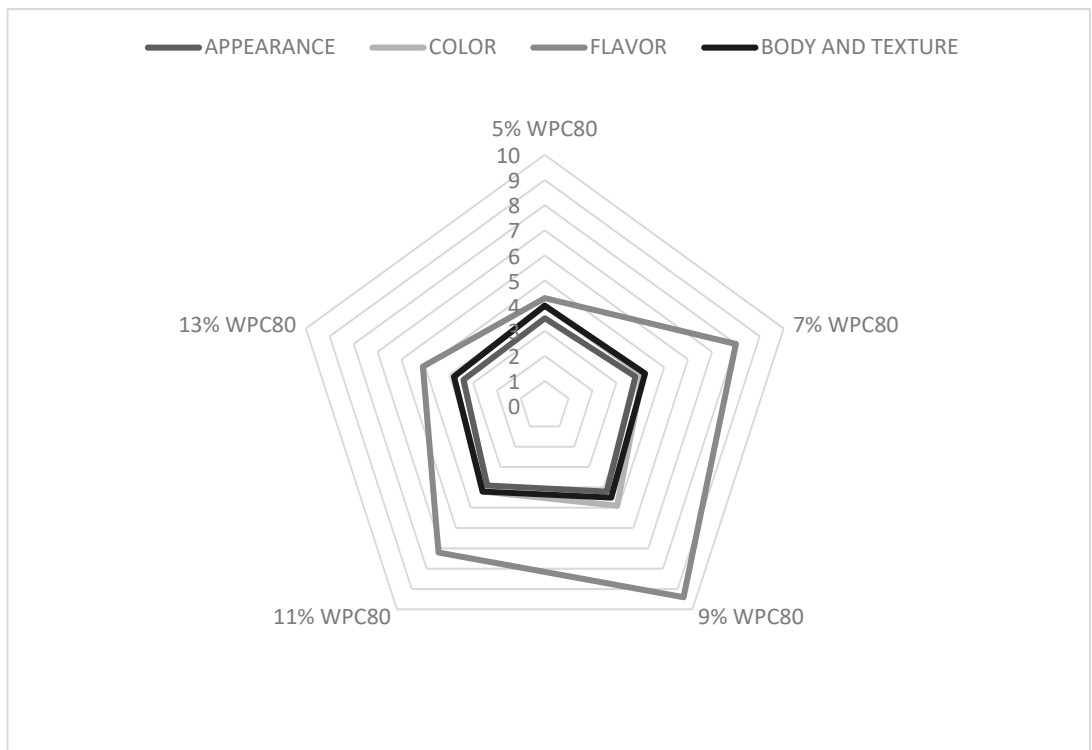

**Figure 6.** Effect of different WPC80 contents on the organoleptic features of high-protein fat-free dairy desserts. Points from 1 to 10 describe the acceptability of the products (Flavor: 1—the lowest score, 10—the highest score; Body, texture, appearance and color: 1—the lowest score, 5—the highest score).

All tested sensory properties of the dessert samples were evaluated after trying the fresh products (stored overnight after manufacturing). Most of the samples were well accepted by the panelists, but differences between each product were noticeable. Each sample was evaluated using a 1–10-point

scale for the description of flavor and a 1–5-point scale for the body, texture, appearance and color of the desserts. The highest point scores related to the greater acceptability of a given feature by panelists. Panelists evaluated the products objectively. Then the scores received from all study participants for each described feature were added up and averaged. The dairy dessert with 13% WPC80 obtained the lowest scores among all tested samples for all tested features (flavor: 5.1, body and texture: 3.8, appearance: 3.4 and color: 3.8). This was associated with its less liquid state and its characteristic flavor and the color of the whey proteins. Opinions from testers confirmed our results regarding the uneven product texture. The highest scores from panelists relating to two of the features were received by the dessert with 9% WPC80 (appearance: 4.2, flavor: 9.4, color: 4.9, body and texture: 4.5). In Figure 6, it can be observed that the product with 9% WPC80 had the highest number of points. It was described as uniform without lumps or air bubbles. The presence of air bubbles and phase separation in the samples with 13% WPC80 was noted and, because of that, this sample received the lowest scores (Figure 6). This product was described as the hardest, with air lumps and bubbles in its structure and with the strongest scent of whey proteins and a yellow color. There is not much literature describing the organoleptic analysis of WPC dairy desserts. Most available publications focus on the relationship between organoleptic and rheological properties. For example, Tárrega and Costell [61], in their research on semi-solid dairy desserts, described the differences between color and thickness of prepared samples. They mentioned that these two features affected the acceptability of the product by consumers. They also noted that the oral thickness of dairy desserts revealed a good correlation to yield stress and apparent viscosity values [61]. We can confirm these observations. Furthermore, our research provides new information on the texture and taste of dairy desserts. The correlation between the amount of added whey protein and the acceptability of the product was noted.

Based on the sensory evaluation, we can conclude that the intensive taste and smell of WPC80 in our product attracted the most attention from people who tested the fat-free dairy desserts. The liquid texture and high viscosity were also not well received. The homogeneous and uniform consistency of the product with 9% WPC80 represents the product that is the closest to the commercial dairy desserts that the typical consumer is used to eating. We conclude that this made this product the highest-rated sample.

## 4. Conclusions

Based on our experiment, it can be concluded that the use of different quantities of whey proteins (5, 7, 9, 11 or 13% WPC80) significantly influenced the textural and rheological properties and appearance of high-protein fat-free dairy desserts in varying degrees. Generally, the increase of WPC80 content increased the hardness (5–11% WPC80 content), adhesiveness (5–13% WPC80 content), cohesiveness (5–13% WPC80 content), springiness (5–11% WPC80 content) and viscosity (5–13% WPC80 content) of the high-protein fat-free dairy desserts tested. A correlation of storage (G′) and loss (G″) moduli, and back-extrusion with hardness values, indicating that the obtained product becomes harder and more compact with the increased addition of WPC80 (5–11%), was noted. This may be the result of interactions between the individual ingredients of the product, for example, carrageenan with proteins through electrostatic or ionic bonding, or by phase separation caused by β-lactoglobulins and k-carrageenan interactions. The tested product showed a significant decrease ($p < 0.05$) in the value of water activity along with an increase in the content of whey-protein concentrate. Every tested sample showed antioxidant properties; however, the sample with 13% WPC80 exhibited the highest value. The dessert with 9% WPC80 was described as the most favorable among panelists in terms of its sensory properties. At the same time, the use of high quantities of whey proteins makes these proteins very noticeable in the product in terms of taste and smell. In addition, the greatest technological problems during the preparation of the samples occurred in the production of the dessert with 13% WPC80, which can be ascribed to the phase separation between whey proteins and k-carrageenan.

The conducted research and the obtained results allow us to conclude that it is possible to use whey proteins as a fat substitute in the preparation of high-protein fat-free dairy desserts. Moreover, these types of products, having health-promoting properties, can benefit consumers.

**Author Contributions:** Conceptualization, B.G.S. and K.K.; methodology, B.G.S. and W.R.; software, B.G.S. and W.R.; validation, B.G.S. and W.R.; formal analysis, K.K.; investigation, K.K.; resources, J.O.S. and B.G.S.; data curation, K.K. and J.O.S.; writing—original draft preparation, K.K. and J.O.S.; writing—review and editing, B.G.S.; visualization, K.K. and J.O.S.; supervision, B.G.S. and W.R.; project administration, B.G.S.; funding acquisition, B.G.S. All authors have read and agreed to the published version of the manuscript.

**Funding:** This research received no external funding.

**Conflicts of Interest:** The authors declare no conflict of interest.

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
