# Peer review of "Effect of Whey Protein Concentrate on Physicochemical, Sensory and Antioxidative Properties of High-Protein Fat-Free Dairy Desserts"

_applsci, doi:10.3390/app10207064_

Round 1

Reviewer 1 Report

Abstract

Please state the type of carrageenan that was used.

It is not clear what the different percentages assigned to the different rheological characteristics of WPC mean. Are these % the increase in the functional properties of the samples or are these the WPC concentrations used as reported in L15.

L16, which experiments are the authors’ refereeing to when they say based on the experiments?

It would be beneficial to the reader if a summary of how the fat free dairy dessert was made. In the current state of the manuscript, this is not clear.

L25-26, what do the authors mean by whey by-product?  And based on what data do the authors claim to have a healthy food? Did the authors determine the caloric content of the dessert? Also given the amount of sucrose used, could this still be termed as being healthy without data e.g. from Nutritics, Alacalc, or any other software?

It is also not clear why the authors have 10 keywords?

Introduction

L31, what does grove mean?

L32, which nowadays trends are the author’s referring to? If anything most people are now being inclined towards plant based/vegan diets due to things like animal welfare, health consciousness. Allergies etc. and dairy may become questionable in this context.

L48-49, the other 20% of whey which is mainly BSA, and Immunoglobulins needs to be mentioned too.

L55 – 69, the authors need to rewrite the information and discuss only caseins then whey and not casein in one sentence and then whey in the next sentence as this is rather confusing to the reader as well as may suggest that whey and casein are being used interchangeably when they are clearly not the same thing.

L70-76, please add references to back up the information that has been provided.

L77-79, please add the type of carrageenan that was used as there are three types and not all of them are able to form gels even in the presence of salts such as calcium and potassium.

In general, the authors need to add the knowledge gap that their study is seeking to fill. There have been studies on fat-free desserts using WPC e.g. Vidigal, Márcia Cristina Teixeira Ribeiro, et al. "Effect of whey protein concentrate on texture of fat-free desserts: sensory and instrumental measurements." Food Science and Technology 32.2 (2012): 412-418. Thus, the authors need to clearly outline in the introduction why they carried out their research and what was the motivation behind this.

Materials and methods

L101, please add the specific type of carrageenan.

L103 what does the mixture mean? Is it carrageenan and water or does this include the other ingredients too?

L104, why were the samples immersed in water at 80 degrees and for how long was this carried out?  And what does the contents mean and why were they homogenised?

L110 – 116 the authors need to account for the effect of the container on the measurements given that carrying out penetration test inside a plastic container there will definitely be an impact from the walls of the container on the measurements.

L117 – 121 at what shear rate was the viscosity measurements carried out?

L123-124 please revise a word is missing in the sentence.

L125, what makes the authors refer to the dessert as high protein? For the % used did the authors check for the solubility to ensure that all of the WPC was soluble in the water to account for all the protein added as WPC usually has lower solubility than WPI.

L139 which samples are the authors referring too? Are the samples the desserts or?

L141-142 at what temperature was the centrifugation carried out?

L150 please add the section where the extraction was carried out.

L159 were the sensory panellist trained or untrained?

L164, see the remark I made earlier on regarding WPC solubility as this could be the main reason for the formation of the protein clots. Additionally depending on the temperature at which the carrageenan was added to the WPC solutions local denaturation could already start to occur and this could impair homogeneity of the samples. However, this should have been resolved by homogenisation hence the authors need to provide some thoughts around this observations.

L167 – 170 please add details in the statistical analysis how the sensory data was analysed.

The authors need to have a control sample. This needs to be either a conventional dessert made using fat or without WPC and then they compare this to fat free dessert to see how it compares to their formulations.

Results and Discussion

Please revise L173 as there are a number of hydrocolloids are basically functional carbohydrates that can disperse in water to result in the formation of gels.

L177 – 179, please rewrite based on the comment provided in the abstract section to be clear what the %s mean.

L187 – 211. The authors discuss how hardness was affected by concentration but do not factor in the possibility of protein solubility. There is a possibility that at this concentration, there samples are not homogenous and would introduce micro-cracks which impair the hardness of the samples. Thus it would be beneficial is the microstructure or at least photos of these samples was included in the manuscript.

L235 – 234, please use a different model system to explain your results as foams cannot be used in this context based on how structurally they differ from the type of samples that you have.

L253 – 265. What is the desirable viscosity of the dessert that the authors are formulating? Higher viscosity means higher resistance to flow is this what the authors would like to have with their product or do they have an optimal value e.g. from literature or model food system to relate the rest of the results to?

L291 – 292, please add reference of other researchers that your findings are similar to.

L304 Please replot figure 2 accordingly.

L315-316 The authors need to discuss their results by providing an argument based on the charge of charge density of carrageenan and how this can interact with WPC based on the charge as well then they can be able to talk about milk proteins and carrageenan interactions.

Figure 3 does not have a Y-axis label.  The aw can also be converted to %.

L340 – 379, the authors need to discuss why these two assays gave completely different results. E.g. with DPPH 11% results are way lower than the same concentration using FRAP. Why is this so?

L383 – 386 prepared products were evaluated by 10 panellists of the staff members of the Faculty of Food Sciences and Biotechnology at the University of Life Sciences in Lublin, Poland Organoleptic evaluation included, appearance, color, texture and flavor of dairy desserts. The presence of air bubbles and protein clots were also considered. The evaluated features of prepared product were as follows: flavor (1–10 points), body, texture, appearance and color (1–5 points) are a repeat of what is stated in the methods section and need to be deleted from the discussion as this is not part of the discussion.

L391-392 what does fresh products mean?

L398 – 399 another reason why microstructure needs to be added to the manuscript.

L408 – 409 is contradictory to what has been discussed under the sensory evaluation as 13% samples had lowest scores which would imply less acceptance and still had high viscosity (Fig. 1) and the authors mention that higher viscosity = better acceptance.

Conclusions

L412, the term structure needs to be related to microstructure which is evidently missing link in the manuscript.

L419 what does the greatest strength mean?

L422 The authors cannot attribute issues with 13% WPC to the foaming properties of whey as in this study what carried out was more or less gelation rather than foaming. These two functional properties of proteins are not the same and neither can they be used interchangeably.

L424, Please revise based on my comment to do with high protein content.

L425, The authors cannot claim well-received organoleptic properties as the size of the panelist was rather low as well as it is not clear whether they were trained or not.

Please indicate any ethics involved in the study as it included human participants.

Author Response

University of Life Sciences in Lublin

Faculty of Food Sciences and Biotechnology

Department of Milk Technology and Hydrocolloids

Skromna 8, 20-704 Lublin

Poland

Phone: +48 81 4623350

Fax:     +48 81 4623345

August 25, 2020,

Dear Reviewer,

Our manuscript entitled “Effect of Whey Protein Concentrate on Physicochemical, Sensory and Antioxidative Properties of High-Protein Fat-Free Dairy Desserts” (Ref. No.: applsci-917245) has been revised and is being re-submitted for publication in Applied Sciences Special Issue "Functional Fermented Milk Products.

We have carefully considered each of the comments and made the appropriate revisions in the manuscript. An itemized list of our responses to each of the comments is included below.

Thank you for your kind attention.

Yours faithfully,

Bartosz Sołowiej

We have corrected our manuscript with regard to Reviewers’ and Editor comments:

Reviewer #1:  green color

Reviewer #2: blue color

If both Reviewers had the same comments – green

Reviewer #1: Review of Manuscript Number: applsci-917245

Title: Effect of Whey Protein Concentrate on Physicochemical, Sensory and Antioxidative Properties of High-Protein Fat-Free Dairy Desserts.

  1. Abstract

Please state the type of carrageenan that was used.

Thank you for your comment. We have used ĸ-carrageenan in our study. Also, we have added this information to the manuscript.

It is not clear what the different percentages assigned to the different rheological characteristics of WPC mean. Are these % the increase in the functional properties of the samples or are these the WPC concentrations used as reported in L15.

Thank you for your comment. In lines (16-17) we have added an information about meaning of “%” that we have used to describe WPC concentrations in our study.

L16, which experiments are the authors’ refereeing to when they say based on the experiments?

Thank you for your comment. We were referring to our experiments but we have decided to remove “Base on the experiments” from the sentence.

It would be beneficial to the reader if a summary of how the fat free dairy dessert was made. In the current state of the manuscript, this is not clear.

Thank you for your comment. In lines 14-17 we have added short information about how the fat-free dairy desserts were made.

L25-26, what do the authors mean by whey by-product? And based on what data do the authors claim to have a healthy food? Did the authors determine the caloric content of the dessert? Also given the amount of sucrose used, could this still be termed as being healthy without data e.g. from Nutritics, Alacalc, or any other software?

Thank you for your comment. In lines 27-28 we have explained the term “whey by-product”. We have based our statement about the health-promoting properties of our desserts on the results of measurements of antioxidant properties and the properties of whey proteins described in the Introduction section.

It is also not clear why the authors have 10 keywords?

Thank you for your comment. According to Applied Sciences Instructions for Authors:

“Keywords: Three to ten pertinent keywords need to be added after the abstract. We recommend that the keywords are specific to the article, yet reasonably common within the subject discipline.”

However, according to Reviewer’s comment we have decided to remove one keyword: viscosity.

  1. Introduction

L31, what does grove mean?

Thank you for your comment. We have changed a word “grove” to more accurate “increase” in line 33.

L32, which nowadays trends are the author’s referring to? If anything most people are now being inclined towards plant based/vegan diets due to things like animal welfare, health consciousness. Allergies etc. and dairy may become questionable in this context.

Thank you for your comment. In lines 33-35 we have explained in more details which contemporary nutritional trends we have described.

L48-49, the other 20% of whey which is mainly BSA, and Immunoglobulins needs to be mentioned too.

Thank you for your comment. In lines 53-55 we have included an information suggested by the Reviewer.

L55 – 69, the authors need to rewrite the information and discuss only caseins then whey and not casein in one sentence and then whey in the next sentence as this is rather confusing to the reader as well as may suggest that whey and casein are being used interchangeably when they are clearly not the same thing.

Thank you for your comment. We have divided information about casein and whey proteins. (lines 48-50 and 59-72).

L70-76, please add references to back up the information that has been provided.

Thank you for your comment. In line 75 we have added a reference to back up the information that has been provided.

L77-79, please add the type of carrageenan that was used as there are three types and not all of them are able to form gels even in the presence of salts such as calcium and potassium.

Thank you for your comment. In lines 80 and 82-83 we have provided additional information about carrageenan that we were referring to in the Introduction section. Also, we have added this information to Abstract and Materials and Methods section.

In general, the authors need to add the knowledge gap that their study is seeking to fill. There have been studies on fat-free desserts using WPC e.g. Vidigal, Márcia Cristina Teixeira Ribeiro, et al. "Effect of whey protein concentrate on texture of fat-free desserts: sensory and instrumental measurements." Food Science and Technology 32.2 (2012): 412-418. Thus, the authors need to clearly outline in the introduction why they carried out their research and what was the motivation behind this.

Thank you for your comment. We have cited the above-mentioned reference and in lines 91-95 we have outlined why we have carried out our research. We also have wanted to note that in the publication mentioned by the Reviewer “Effect of whey protein concentrate on texture of fat-free desserts: sensory and instrumental measurements” authors described only few features of prepared dairy desserts. Moreover, they have tested low-protein whey powder (WPC35 – with high lactose content) in low concentrations (1.5-4.5%), contrary to our study. Our research is more complex and provide new information to general knowledge about fat-free dairy desserts, e.g. antioxidative properties, oscillatory rheometry and water activity.

  1. Materials and methods

L101, please add the specific type of carrageenan.

Thank you for your comment. We have added the type of carrageenan (lines 104, 109, 113 and elsewhere).

L103 what does the mixture mean? Is it carrageenan and water or does this include the other ingredients too?

Thank you for your comment. We have specified this information in lines 113-115.

L104, why were the samples immersed in water at 80 degrees and for how long was this carried out? And what does the contents mean and why were they homogenised?

Thank you for your comment. We have included additional information describing time and reason of immersion of samples in water bath in lines 113-115. Moreover, samples were mixed at 10 000 rpm for 10 min. The name of the equipment is “H 500 homogenizer”.

L110 – 116 the authors need to account for the effect of the container on the measurements given that carrying out penetration test inside a plastic container there will definitely be an impact from the walls of the container on the measurements.

Thank you for your comment. We have taken into consideration the effect of plastic container, therefore the test was a modified TPA/penetration test (applied to different semi-solid dairy products e.g. processed cheese, cheese sauces, desserts etc.):

Fox, P. F., McSweeney, P. L. H., Cogan, T. M., & Guinee, T. P. (2017). Fundamentals of cheese science (2nd. ed.). New York: Springer, (Chapter 14).

Szafrańska, J.O.; Muszyński, S.; Sołowiej B.G. Effect of whey protein concentrate on physicochemical properties of acid casein processed cheese sauces obtained with coconut oil or anhydrous milk fat. LWT - Food Sci. Techn. 2020, 127, 109434.

Moreover, in our research, we have used samples in a cylindrical container  (40 mm in diameter) and a cylindrical probe (15 mm), so the forces were distributed evenly.

L117 – 121 at what shear rate was the viscosity measurements carried out?

Thank you for your comment.

The shear rate (spindle velocity in the case of the Helipath Stand and T-bar spindle) was constant at 0.5 rpm. Usually, one spindle velocity is used to determine the Helipath viscosity and there is no defined shear rate.

A fragment from Brookfield Laboratory employee Dr. Moonay (Brookfield Engineering Laboratories, Inc., Middleboro, MA, United States) publication about the use of, inter alia, the Helipath Stand with T-bar spindle.

"This technique provides very useful correlations to process engineering data for certain rheologically complex products. However, there are no defined shear rates. … The Helipath and T-bar spindles are suitable if all they need is a reproducible viscosity number without shear rate." "…Therefore, many operators use the Helipath™ Stand Model system (Brookfield Engineering Laboratories, Inc.), in which a motorized stand lowers a viscometer head while a thin T-shaped spindle rotates at a low to moderate speed, such as 10 rpm or less." - Moonay, D. J. (2012). Viscosity measurements as a sensitive method for product shelf-life testing. American Laboratory, 44(8), 10-15.

The analysis of Helipath viscosity is also described in: Skriver, A., Holstborg, J., & Qvist, K. B. (1999). Relation between sensory texture analysis and rheological properties of stirred yogurt. Journal of Dairy Research, 66(4), 609-618. "…More empirical measures of viscosity were obtained by measuring the outlet time for a standard volume to pass through a Posthumus funnel (orifice, 8 mm; measuring temperature, 5°C) and using a Brookfield viscometer (model DV II; Brookfield Engineering Laboratories, Stoughton, MA 02072, USA) with Helipath stand, a type D T-bar spindle and a speed of 5 rev./min."

L123-124 please revise a word is missing in the sentence.

Thank you for your comment. We have corrected the sentence (line 136).

L125, what makes the authors refer to the dessert as high protein? For the % used did the authors check for the solubility to ensure that all of the WPC was soluble in the water to account for all the protein added as WPC usually has lower solubility than WPI.

Thank you for your comment. According to publication (P.J.Luck, B.Vardhanabhuti, Y.H.Yong, T.Laundon, D.M.Barbano, E.A.Foegeding, Journal of Dairy Science, Volume 96, Issue 9, September 2013, Pages 5522-5531 “Comparison of functional properties of 34% and 80% whey protein and milk serum protein concentrates”) WPC80 is generally highly soluble in water. Only in pH below 3.0 (which does not apply to dairy desserts) the solubility of WPC80 is lower. Moreover, WPC80 was dissolved in water in room temperature.

We have used the term “high-protein dairy desserts”, because most of dairy desserts contain less than 10% of total protein (usually 7-8%): e.g. Zarzycki et al. (“Rheological properties of milk-based desserts with the addition of oat gum and kappa-carrageenan” J Food Sci Technol., 2019, 56(11):5107–5115). We have added from 5 to 13% of WPC80, not counting SMP.

L139 which samples are the authors referring too? Are the samples the desserts or?

Thank you for your comment. We have clarified this in line 152 “1 g of every dessert”.

L141-142 at what temperature was the centrifugation carried out?

Thank you for your comment. The samples were centrifuged at 4 °C. We have added this information - line 155).

L150 please add the section where the extraction was carried out.

Thank you for your comment. We put all information on extraction in a different section, as suggested (lines 151-155).

L159 were the sensory panellist trained or untrained?

Thank you for your comment. Panelists were untrained. We have added additional information about panelists in line 176.

L164, see the remark I made earlier on regarding WPC solubility as this could be the main reason for the formation of the protein clots. Additionally depending on the temperature at which the carrageenan was added to the WPC solutions local denaturation could already start to occur and this could impair homogeneity of the samples. However, this should have been resolved by homogenisation hence the authors need to provide some thoughts around this observations.

Thank you for your comment. We have added an explanation to the Results and Discussion section.

L167 – 170 please add details in the statistical analysis how the sensory data was analysed.

Thank you for your comment. The results from sensory analysis were not statistically analyzed, similar to other publication: SzafraÅ„ska, J.O.; SoÅ‚owiej, B.G. Effect of different fibres on texture, rheological and sensory properties of acid casein processed cheese sauces. Int. J. Food Sci. Tech. 2020, 55, 1971–1979.

The authors need to have a control sample. This needs to be either a conventional dessert made using fat or without WPC and then they compare this to fat free dessert to see how it compares to their formulations.

Thank you for your comment. We agree, that normally we should compare results to control sample as you mentioned before, but the aim of our project was to compare the samples with each other in order to choose the best composition (base) for further research.

  1. Results and Discussion

Please revise L173 as there are a number of hydrocolloids are basically functional carbohydrates that can disperse in water to result in the formation of gels.

Thank you for your comment. We have decided to remove this sentence.

L177 – 179, please rewrite based on the comment provided in the abstract section to be clear what the %s mean.

Thank you for your comment. In line 204 we explain in more details what “%” means.

L187 – 211. The authors discuss how hardness was affected by concentration but do not factor in the possibility of protein solubility. There is a possibility that at this concentration, there samples are not homogenous and would introduce micro-cracks which impair the hardness of the samples. Thus it would be beneficial is the microstructure or at least photos of these samples was included in the manuscript.

Thank you for your comment. We have provided an explanation with regard to decrease in hardness (13% WPC80) (lines 231-235). Also, we agree with the Reviewer with regard to microstructure. Product microstructure studies will be performed in the next step of the whole project. Moreover, in our opinion, the article contains quite big amount of methods and results, and about 8000 words. Another measurements would make the article much longer, usually too long for one publication.

L235 – 234, please use a different model system to explain your results as foams cannot be used in this context based on how structurally they differ from the type of samples that you have.

Thank you for your comment. In lines 272-277 we have edited a whole fragment of the text and we have used different model system to describe this topic. Also, we have added new reference: Antczak, M.; Pluta, A.; Lenart, A.; Berthold-Pluta, A. The effect of heat treatment temperature and active acidity on textural properties of whey protein-pectin gels. Zeszyty Problemowe PostÄ™pów Nauk Rolniczych, 2018, 595, 3–11. DOI 10.22630/ZPPNR.2018.595.30

L253 – 265. What is the desirable viscosity of the dessert that the authors are formulating? Higher viscosity means higher resistance to flow is this what the authors would like to have with their product or do they have an optimal value e.g. from literature or model food system to relate the rest of the results to?

Thank you for your comment. In lines 282-285 we have added an information about desirable viscosity of the dessert.

L291 – 292, please add reference of other researchers that your findings are similar to.

Thank you for your comment. In line 331 we have added a reference.

L304 Please replot figure 2 accordingly.

Thank you for your comment. We have corrected Figure 2 (lines 342-343).

L315-316 The authors need to discuss their results by providing an argument based on the charge of charge density of carrageenan and how this can interact with WPC based on the charge as well then they can be able to talk about milk proteins and carrageenan interactions.

Thank you for your comment. We have added suggested information in lines 353-358. New references: McHugh, D. J. A guide to the seaweed industry. FAO Fisheries technical paper http://www.fao.org/3/y4765e/y4765e0a.htm Access 23.08.2020

Figure 3 does not have a Y-axis label. The aw can also be converted to %.

Thank you for your comment. We have corrected Figure 3 by adding Y-axis label (lines 371-373) With regard to aw, we have decided to keep the values.

L340 – 379, the authors need to discuss why these two assays gave completely different results. E.g. with DPPH 11% results are way lower than the same concentration using FRAP. Why is this so?

Thank you for your comment. We have added an explanation to the manuscript (lines 426-430). Different antioxidant methods differ in chemical mechanism and may give sometimes different results. However, in the present research the data on DPPH and FRAP method correlated well (correlation coefficient, R=0.84). Some discrepancies between the data from FRAP and DPPH antioxidant methods are not uncommon and occurred before with different biological material (Radzki et al., 2014; Thaipong et al. 2006;  Djordjevic et al. 2011).

Radzki W, SÅ‚awiÅ„ska A, JabÅ‚oÅ„ska-RyÅ› E, Gustaw W. Antioxidant capacity and polyphenolic content of dried wild edible mushrooms from Poland. Int J Med Mushrooms. 2014;16(1):65–75.

Thaipong K, Boonprakob U, Crosby K, Cisneros-Zevallos L, Hawkins Byrne D. Comparison of ABTS, DPPH, FRAP, and ORAC assays for estimating antioxidant activity from guava fruit extracts. J Food Compos Anal. 2006;19(6–7):669–75.

Djordjevic TM, Šiler-Marinkovic SS, Dimitrijevic-Brankovic SI. Antioxidant activity and total phenolic content in some cereals and legumes. Int J Food Prop. 2011;14(1):175–84.

L383 – 386 prepared products were evaluated by 10 panellists of the staff members of the Faculty of Food Sciences and Biotechnology at the University of Life Sciences in Lublin, Poland Organoleptic evaluation included, appearance, color, texture and flavor of dairy desserts. The presence of air bubbles and protein clots were also considered. The evaluated features of prepared product were as follows: flavor (1–10 points), body, texture, appearance and color (1–5 points) are a repeat of what is stated in the methods section and need to be deleted from the discussion as this is not part of the discussion.

Thank you for your comment. We have deleted repeated information in the text.

L391-392 what does fresh products mean?

Thank you for your comment. We have added an explanation in line 438.

L398 – 399 another reason why microstructure needs to be added to the manuscript.

Thank you for your comment. Also, we agree with the Reviewer with regard to microstructure. Product microstructure studies will be performed in the next step as part of the whole project. Moreover, in our opinion, the article contains quite big amount of methods and results, and about 8000 words. Another measurements would make the article much longer, usually too long for one publication.

L408 – 409 is contradictory to what has been discussed under the sensory evaluation as 13% samples had lowest scores which would imply less acceptance and still had high viscosity (Fig. 1) and the authors mention that higher viscosity = better acceptance.

Thank you for your comment. We have decided to remove this sentence.

  1. Conclusions

L412, the term structure needs to be related to microstructure which is evidently missing link in the manuscript.

Thank you for your comment. We have changed the word “structure” to “textural and rheological properties”, which we have evaluated in our study (line 457).

L419 what does the greatest strength mean?

Thank you for your comment. We have reformulated the sentence in line 464.

L422 The authors cannot attribute issues with 13% WPC to the foaming properties of whey as in this study what carried out was more or less gelation rather than foaming. These two functional properties of proteins are not the same and neither can they be used interchangeably.

Thank you for your comment. We have reformulated the explanation and deleted unnecessary information (467-468).

L424, Please revise based on my comment to do with high protein content.

Thank you for your comment. We have revised this sentence 466-468.

L425, The authors cannot claim well-received organoleptic properties as the size of the panelist was rather low as well as it is not clear whether they were trained or not.

Thank you for your comment. We have removed this sentence.

Please indicate any ethics involved in the study as it included human participants.

Thank you for your comment. We have added appropriate statement (lines 183-194).

Thank you very much.

Reviewer 2 Report

The authors sought to explore the physicochemical, sensory and antioxidative properties of high-protein fat-free dairy desserts. I felt that a lot of effort was contributed to achieve this goal, and this manuscript is clear logic and well-written. However, it is mandatory to explain/correct the manuscript in some points.  
1. The composition of fat free dairy dessert should be provided.
2. Line 116. Authors claimed three replicates. Does it mean analytical or biological replicate?
3. Please add the specific type of carrageenan in the material part.

Author Response

University of Life Sciences in Lublin

Faculty of Food Sciences and Biotechnology

Department of Milk Technology and Hydrocolloids

Skromna 8, 20-704 Lublin

Poland

Phone: +48 81 4623350

Fax:     +48 81 4623345

August 25, 2020,

Dear Reviewer,

Our manuscript entitled “Effect of Whey Protein Concentrate on Physicochemical, Sensory and Antioxidative Properties of High-Protein Fat-Free Dairy Desserts” (Ref. No.: applsci-917245) has been revised and is being re-submitted for publication in Applied Sciences Special Issue "Functional Fermented Milk Products.

We have carefully considered each of the comments and made the appropriate revisions in the manuscript. An itemized list of our responses to each of the comments is included below.

Thank you for your kind attention.

Yours faithfully,

Bartosz Sołowiej

We have corrected our manuscript with regard to Reviewers’ and Editor comments:

Reviewer #1:  green color

Reviewer #2: blue color

If both Reviewers had the same comments – green

Reviewer #2: Review of Manuscript Number: applsci-917245

Title: Effect of Whey Protein Concentrate on Physicochemical, Sensory and Antioxidative Properties of High-Protein Fat-Free Dairy Desserts.

The authors sought to explore the physicochemical, sensory and antioxidative properties of high-protein fat-free dairy desserts. I felt that a lot of effort was contributed to achieve this goal, and this manuscript is clear logic and well-written. However, it is mandatory to explain/correct the manuscript in some points.

Thank you very much for your opinion about the manuscript.

  1. The composition of fat free dairy dessert should be provided.

Thank you for your comment. We have provided the composition of fat-free dairy dessert (lines 121-122).

  1. Line 116. Authors claimed three replicates. Does it mean analytical or biological replicate?

Thank you for your comment. Analytical. “Three replicates” means that we were obtaining dairy desserts three times. After that, we were performing five measurements (penetration tests) to each of the batch.

  1. Please add the specific type of carrageenan in the material part.

Thank you for your comment. We have provided additional information about carrageenan (ĸ-carrageenan) that we were referring to in the Introduction section. Also, we have added this information to Abstract and Materials and Methods section.

Thank you very much.

Round 2

Reviewer 1 Report

The authors have worked on revising the manuscript swiftly. However, I am afraid that the revised manuscript is not yet to the level of being accepted for publication. Please see below some comments that need to be addressed:-

L17 – 19, am not very sure why the conclusion has been added at this point.

L34-35 needs to be revised as it is not clear.

L53-54 please revise

L67-68 is not clear. Who does they refer to? And what resulted in inhibitory activity against Helicobacter pylori? Also Okuda et al. in a study upon bovine lactoferrin statement needs to be rewritten.

L61 Unfortunately most of the statements in L 61 – l70 have been lifted from the article by Davoodi, S. H., Shahbazi, R., Esmaeili, S., Sohrabvandi, S., Mortazavian, A., Jazayeri, S., & Taslimi, A. (2016). Health-Related Aspects of Milk Proteins. Iranian journal of pharmaceutical research: IJPR, 15(3), 573–591. This is not acceptable as either the statements need to be paraphrased or quoted.

L93 – 95 why are these parameters important for one to study?

L234, I agree with the statement written by the authors and would like to know what was the pH of the final mixtures in order for this statement to be applicable in their case.

L282 – 285 is not clear and needs to be revised.

L353 – 360 this whole paragraph is all confused and mixed up. It’s true that kappa carrageenan is negatively charged and will interact with molecules such as potassium. However, in the case of the current research, did the authors add potassium to their formulations in order for them to relate to this explanation or were they working with protein?

L426 – 430 why is this so?

L437 – 454 the radar graph is correctly plotted. However, the results needs to be revised to give a clear interpretation of the same and reflect on the data provided in the radar graph.

Additional comments on how L437 -454 needs to be revised can be seen below:-

L438 – 439 the information needs to be elaborated on to make it clear what you mean

L443 please expand on how you determined the differences in the consistency? Did you use the power law which is the optimal way that is recognised to record quantitative consistency data?

L433 please elaborate on what the highest notes mean in this context.

L445 please specify which product you are referring to.

This previous comment needs to be addressed in the manuscript. L110 – 116 the authors need to account for the effect of the container on the measurements given that carrying out penetration test inside a plastic container there will definitely be an impact from the walls of the container on the measurements.

 I appreciate the fact that the authors mention that “We have taken into consideration the effect of plastic container, therefore the test was a modified TPA/penetration test (applied to different semi-solid dairy products e.g. processed cheese, cheese sauces, desserts etc.)” but this information needs to be included in the manuscript.

Author Response

University of Life Sciences in Lublin

Faculty of Food Sciences and Biotechnology

Department of Milk Technology and Hydrocolloids

Skromna 8, 20-704 Lublin

Poland

Phone: +48 81 4623350

Fax:     +48 81 4623345

September 05, 2020,

Dear Reviewer,

Our manuscript entitled “Effect of Whey Protein Concentrate on Physicochemical, Sensory and Antioxidative Properties of High-Protein Fat-Free Dairy Desserts” (Ref. No.: applsci-917245 – Round 2) has been revised and is being re-submitted for publication in Applied Sciences.

We have carefully considered each of the comments and made the appropriate revisions in the manuscript. An itemized list of our responses to each of the comments is included below.

Thank you for your kind attention.

Yours faithfully,

Bartosz Sołowiej

We have corrected our manuscript with regard to Reviewers’ comments:

Reviewer #1:  green color

Reviewer #1: Review of Manuscript Number: applsci-917245 (Round 2)

Title: Effect of Whey Protein Concentrate on Physicochemical, Sensory and Antioxidative Properties of High-Protein Fat-Free Dairy Desserts.

Abstract

L17 – 19, am not very sure why the conclusion has been added at this point.

Thank you for your comment. The sentence has been modified. Moreover, we have added it to introduce the reader to the fragment of the text where authors explain the occurred changes of the desserts’ properties (lines 17-19).

Introduction

L34-35 needs to be revised as it is not clear.

Thank you for your comment. We have reformulated the whole fragment (lines 33-37). Now, should be clear. 

L53-54 please revise

Thank you for your comment. We have rewritten the sentence (lines 52-53).

L67-68 is not clear. Who does they refer to? And what resulted in inhibitory activity against Helicobacter pylori? Also Okuda et al. in a study upon bovine lactoferrin statement needs to be rewritten.

Thank you for your comment. We have improved and rewritten the above-mentioned sentences (lines 64-68).

L61 Unfortunately most of the statements in L 61 – l70 have been lifted from the article by Davoodi, S. H., Shahbazi, R., Esmaeili, S., Sohrabvandi, S., Mortazavian, A., Jazayeri, S., & Taslimi, A. (2016). Health-Related Aspects of Milk Proteins. Iranian journal of pharmaceutical research: IJPR, 15(3), 573–591. This is not acceptable as either the statements need to be paraphrased or quoted.

Thank you for your comment. We have improved and rewritten the above-mentioned sentences (lines 62-68).

L93 – 95 why are these parameters important for one to study?

Thank you for your comment. These parameters are important in food product development (rheological properties – G′, G″ etc.), for enhancement of health-promoting properties of food products (antioxidative properties), food processing and storage (water activity).

Results and Discussion

L234, I agree with the statement written by the authors and would like to know what was the pH of the final mixtures in order for this statement to be applicable in their case.

Thank you for your comment. The pH of dairy desserts was 6.2. We have added this information to the 2.2. Preparation of High-Protein Fat-Free Dairy Desserts in “Materials and Methods” section (lines 111-112).

L282 – 285 is not clear and needs to be revised.

Thank you for your comment. We have revised the whole fragment in lines 284-287.

L353 – 360 this whole paragraph is all confused and mixed up. It’s true that kappa carrageenan is negatively charged and will interact with molecules such as potassium. However, in the case of the current research, did the authors add potassium to their formulations in order for them to relate to this explanation or were they working with protein?

Thank you for your comment. In lines 356-359 we have rewritten and improved this part. Also, we did not add potassium, but WPC80 contains potassium in its composition – about 520 mg.

L426 – 430 why is this so?

Thank you for your comment. We have changed this fragment of the manuscript and provided more detailed explanation, hoping that the issue is clarified enough (lines 427-435).

BEFORE

“Different antioxidant methods differ in chemical mechanism and may give sometimes different results. However, in the present research the data on DPPH and FRAP method correlated well (correlation coefficient, R=0.84). Some discrepancies between the data from FRAP and DPPH antioxidant methods are not uncommon and occurred before with different biological material [53-55].”

AFTER

“In the present research the data on DPPH and FRAP method correlated well (correlation coefficient, R=0.84) which is in agreement with many previous studies [53–55]. Some discrepancies between the data from FRAP and DPPH antioxidant methods occurred. However, such discrepancies are generally not uncommon and were reported before with different biological material [55–57]. Moreover, there are reports in which there is no correlation or it is weak [58]. Although both FRAP and DPPH assays are based on transferring of electrons from the antioxidant and reduction an oxidant, FRAP method is more prone to colour interference and slow colour development. It also involves more antioxidant compounds which reacts with various kinetics [59,60].”

  1. Radzki, W.; SÅ‚awinska, A.; JabÅ‚onska-RyÅ›, E.; Michalak-Majewska, M. Effect of blanching and cooking on antioxidant capacity of cultivated edible mushrooms: A comparative study. Int. Food Res. J. 2016, 23, 599–605.
  2. Wanna, C. Free radical scavenging capacity and total phenolic contents in peel and fleshy crude extracts of selected vegetables. Pharmacogn. J. 2019, 11, 1351–1358, doi:10.5530/pj.2019.11.209.
  3. Thaipong, K.; Boonprakob, U.; Crosby, K.; Cisneros-Zevallos, L.; Hawkins Byrne, D. Comparison of ABTS, DPPH, FRAP, and ORAC assays for estimating antioxidant activity from guava fruit extracts. J. Food Compos. Anal. 2006, 19, 669–675, doi:10.1016/j.jfca.2006.01.003.
  4. Radzki, W.; SÅ‚awiÅ„ska, A.; JabÅ‚oÅ„ska-RyÅ›, E.; Gustaw, W. Antioxidant capacity and polyphenolic content of dried wild edible mushrooms from Poland. Int. J. Med. Mushrooms 2014, 16, 65–75, doi:10.1615/IntJMedMushr.v16.i1.60.
  5. Djordjevic, T.M.; Šiler-Marinkovic, S.S.; Dimitrijevic-Brankovic, S.I. Antioxidant activity and total phenolic content in some cereals and legumes. Int. J. Food Prop. 2011, 14, 175–184, doi:10.1080/10942910903160364.
  6. Gil, M.I.; Tomás-Barberán, F.A.; Hess-Pierce, B.; Kader, A.A. Antioxidant capacities, phenolic compounds, carotenoids, and vitamin C contents of nectarine, peach, and plum cultivars from California. J. Agric. Food Chem. 2002, 50, 4976–4982, doi:10.1021/jf020136b.
  7. Clarke, G.; Ting, K.N.; Wiart, C.; Fry, J. High correlation of 2,2-diphenyl-1-picrylhydrazyl (DPPH) radical scavenging, ferric reducing activity potential and total phenolics content indicates redundancy in use of all three assays to screen for antioxidant activity of extracts of plants from the malaysian rainforest. Antioxidants 2013, 2, 1–10, doi:10.3390/antiox2010001.
  8. Huang, D.; Boxin, O.U.; Prior, R.L. The chemistry behind antioxidant capacity assays. J. Agric. Food Chem. 2005, 53, 1841–1856, doi:10.1021/jf030723c.

L437 – 454 the radar graph is correctly plotted. However, the results needs to be revised to give a clear interpretation of the same and reflect on the data provided in the radar graph.

Additional comments on how L437 -454 needs to be revised can be seen below:

L438 – 439 the information needs to be elaborated on to make it clear what you mean

Thank you for your comment. In lines 441-442, 445-448, and 453-454 we have elaborated explanation about interpretation of provided results.

L443 please expand on how you determined the differences in the consistency? Did you use the power law which is the optimal way that is recognised to record quantitative consistency data?

Thank you for your comment. We have meant the texture and body (we have left only texture in the sentence), and therefore we have decided to remove the word “consistency” to be more strict and not to mix up with e.g. the consistency index.

L443 please elaborate on what the highest notes mean in this context.

Thank you for your comment. In lines 447-448 and in the description of radial graph we have added information about points (notes) and their meanings.

L445 please specify which product you are referring to.

Thank you for your comment. We have added an information in line 453.

This previous comment needs to be addressed in the manuscript. L110 – 116 the authors need to account for the effect of the container on the measurements given that carrying out penetration test inside a plastic container there will definitely be an impact from the walls of the container on the measurements.

I appreciate the fact that the authors mention that “We have taken into consideration the effect of plastic container, therefore the test was a modified TPA/penetration test (applied to different semi-solid dairy products e.g. processed cheese, cheese sauces, desserts etc.)” but this information needs to be included in the manuscript.

Thank you for your comment. We have added this information to the Results and discussion section (lines 203-206).

Round 3

Reviewer 1 Report

The authors have submitted a revised version of the manuscript. However, some of the comments that I made have not been sufficiently addressed in the manuscript despite the authors stating that they have made the amendments in the cover letter. A typical example is the sensory data where they say they have elaborated on the information. However, this is not reflected in the manuscript. There are also numerous grammatical errors in the manuscript that need to be revised. Can the authors please consider an in-depth reflection on the review comments provided and ensure that they make significant changes to the version of the paper that they will be resubmitting? Thank you.

Author Response

University of Life Sciences in Lublin

Faculty of Food Sciences and Biotechnology

Department of Milk Technology and Hydrocolloids

Skromna 8, 20-704 Lublin, Poland

Phone: +48 81 4623350

Fax:     +48 81 4623345

September 25, 2020,

Dear Reviewer,

Our manuscript entitled “Effect of Whey Protein Concentrate on Physicochemical, Sensory and Antioxidative Properties of High-Protein Fat-Free Dairy Desserts” (Ref. No.: applsci-917245 – Round 3) has been revised and is being re-submitted for publication in Applied Sciences.

We have carefully considered each of the comments and made the appropriate revisions in the manuscript (3 rounds – 3 colors – yellow, blue and grey). All the additional improvements (Round 3) have been marked in grey.

An itemized list of our responses to each of the comments is included below and in the manuscript.

Moreover, our manuscript has been corrected by an American native speaker (track changes).

Thank you for your kind attention.

Yours faithfully,

Bartosz Sołowiej

We have corrected our manuscript with regard to Reviewers’ comments:

Reviewer #1:  grey color (Round 3)

Reviewer #1: Review of Manuscript Number: applsci-917245 (Round 3)

Title: Effect of Whey Protein Concentrate on Physicochemical, Sensory and Antioxidative Properties of High-Protein Fat-Free Dairy Desserts.

The authors have submitted a revised version of the manuscript. However, some of the comments that I made have not been sufficiently addressed in the manuscript despite the authors stating that they have made the amendments in the cover letter.

A typical example is the sensory data where they say they have elaborated on the information. However, this is not reflected in the manuscript.

There are also numerous grammatical errors in the manuscript that need to be revised.

Can the authors please consider an in-depth reflection on the review comments provided and ensure that they make significant changes to the version of the paper that they will be resubmitting? Thank you.

Thank you for your comment. We have addressed all the issues in our manuscript, but to clarify main topics and better discuss our experiments, we have added new information. All the additional improvements (Round 3) have been marked in grey in the manuscript.

An itemized list of our responses to each of your comments is included below and in the manuscript (3 rounds – 3 colors – yellow, blue and grey).

Moreover, our manuscript has been corrected by American Native Speaker (track changes). Errors were corrected.

Abstract

L17 – 19, am not very sure why the conclusion has been added at this point.

Thank you for your comment. The sentence has been modified. Moreover, we have added it to introduce the reader to the fragment of the text where authors explain the occurred changes of the desserts’ properties (lines 18-19).

BEFORE:

It can be concluded that the use of different amounts of whey proteins significantly influenced the texture, rheological properties and appearance of dairy desserts.

AFTER:

The use of different amounts of whey proteins significantly influenced the texture, rheological properties and appearance of dairy desserts.

Introduction

L34-35 needs to be revised as it is not clear.

Thank you for your comment. We have reformulated the whole fragment (lines 34-39). Now, should be clear. 

BEFORE:

Dairy desserts market is expected to increase during upcoming few years. This type of products is abundant in calcium, potassium and vitamins, which fits nowadays nutritional trends searching for food containing health-promoting components. Rising demand for sweet and healthy dishes is connected with a growth of the global dairy desserts market. It is worth to mention that growing interest in new dairy products forces producers to invent new formulas and innovations [1].

AFTER:

The dairy desserts market is expected to grow in the next few years thanks to new nutritional trends searching for ready-to-eat food containing health-promoting components that are also low in fat. This type of product is abundant in calcium, potassium and vitamins, which fits well with the demand for health-promoting, low-fat desserts. Growing interest in new dairy products forces producers to invent new formulas and innovations [1].

L53-54 please revise

Thank you for your comment. We have rewritten the sentence (line 55-56).

BEFORE:

Proteins constitute 0.6-0.8% of the whey mass, of which 70-80% are α-lactalbumin and β-lactoglobulin.

AFTER:

70–80% of all proteins found in whey mass are α-lactalbumin and β-lactoglobulin.

L67-68 is not clear. Who does they refer to? And what resulted in inhibitory activity against Helicobacter pylori? Also Okuda et al. in a study upon bovine lactoferrin statement needs to be rewritten. L61 Unfortunately most of the statements in L 61 – L70 have been lifted from the article by Davoodi, S. H., Shahbazi, R., Esmaeili, S., Sohrabvandi, S., Mortazavian, A., Jazayeri, S., & Taslimi, A. (2016). Health-Related Aspects of Milk Proteins. Iranian journal of pharmaceutical research: IJPR, 15(3), 573–591. This is not acceptable as either the statements need to be paraphrased or quoted.

Thank you for your comment. We have improved and rewritten the above-mentioned sentences (lines 64-73).

BEFORE:

New research also confirms that milk proteins, especially whey proteins, may protect the human body against different types of cancers like colon, breast, and prostate gland. This ability is associated with enhancement of the cellular levels of glutathione and promotion of hormonal and cell-mediated immune responses [8, 11]. Several in vitro and in vivo research had proven that whey proteins can positively influence the human immune responses [8]. Also, researchers have confirmed that whey proteins contain a wide range of unique components with antimicrobial activity. They noticed inhibitory activity against Helicobacter pylori in infected subjects. Okuda et al. in a study upon bovine lactoferrin as effective suppressant of Helicobacter pylori colonization in the human stomach, revealed that twice daily oral administration of lactoferrin tablets for 12 weeks decreased the ability of H. pylori to form colonies [12]. Moreover, milk proteins are important additives in dairy industry production due to their properties such as solubility, high nutritional value, bland flavor profile and properties to reduce the fat content [13].

AFTER:

It has also been confirmed that milk proteins, especially whey proteins, may have an influence on the protection of the human body against different types of cancer, such as breast, colon or prostate gland cancer. This ability is associated with the increase of cellular glutathione levels and the promotion of hormonal and cell-mediated immune reactions [8, 11]. Davoodi et al. [8] reported that several in vitro and in vivo studies had found evidence that whey proteins could positively impact human immune responses. They observed the inhibitory activity of whey proteins against Helicobacter pylori in infected subjects. Furthermore, Okuda et al. [12] found that the oral administration of lactoferrin tablets taken twice a day for 12 weeks reduced the capacity of H. pylori to form colonies.

L93 – 95 why are these parameters important for one to study?

Thank you for your comment.

We have added above information to main text (lines 103-105):

“These parameters are important in food product development (rheological properties – G′, G″, etc.) for the enhancement of the health-promoting properties of food products (antioxidative properties), food processing and storage (water activity).”

ADDITIONALLY

We have improved an “Introduction” section by adding new information to the text.

In lines 76-80 you can find additional sentences:

Reis et al. [6] have suggested that the consumption of products containing saturated fat is connected with heart disease. Therefore, there is a need for new food systems either containing lower amounts of fat or that are fat free [6]. From a technological point of view, fat is an important ingredient that affects the taste of a product, but also improves the texture and mouthfeel.

Results and Discussion

L234, I agree with the statement written by the authors and would like to know what was the pH of the final mixtures in order for this statement to be applicable in their case.

Thank you for your comment. The pH of dairy desserts was 6.2. We have added this information to the 2.2. Preparation of High-Protein Fat-Free Dairy Desserts in “Materials and Methods” section (lines 123-124).

BEFORE

“…at room temperature, and then heated for 15 min at approx. 70 °C while stirring. Then the mixture…”

AFTER

“Then the pH of the WPC80, SMP, sucrose and ĸ-carrageenan mixture was adjusted to 6.2 using citric acid or sodium hydroxide (2 mol/l). After that, …”

We also have added extended explanation of our results in lines 253-256 and 258-262, respectively:

“Thermodynamic incompatibility (phase separation) is the most common phenomenon occurring between proteins and hydrocolloids in a system where pH surpasses the protein isoelectric point [26-27], which is in line with our study where the pH was 6.2.”

“They also noticed that in a product with added coconut oil, not only were the values of hardness lower in samples with the highest amount of added whey protein concentrate compared to a product with a lower amount of WPC80, but also the adhesiveness of the samples. Adhesiveness is a force that is required to remove food from the palate or teeth while chewing [23].”

L282 – 285 is not clear and needs to be revised.

Thank you for your comment. We have revised the whole fragment (lines 309-312).

BEFORE

Based on observations of the tested products, dairy dessert with the highest and the lowest values of tested features did not resemble the texture of commercial dairy desserts, to which we want to make our product similar.

AFTER

The obtained results and visual observations of our tested products allowed us to conclude that the highest and the lowest values of the tested feature (viscosity) did not resemble the texture of any of the commercial dairy desserts that we wanted our product to be similar to.

L353 – 360 this whole paragraph is all confused and mixed up. It’s true that kappa carrageenan is negatively charged and will interact with molecules such as potassium. However, in the case of the current research, did the authors add potassium to their formulations in order for them to relate to this explanation or were they working with protein?

Thank you for your comment. In lines 384-391 we have rewritten and improved this part. Also, we did not add potassium, but WPC80 contains potassium in its composition – about 520 mg.

BEFORE

They are enabling to interact with positively charged proteins. For example, carrageenan combines with the milk protein like casein and can form a three-dimensional gel network [43]. Gustaw et al. [2] suggested that carrageenan affects the increase in hardness of dairy desserts, but at higher concentration, it may cause adverse changes in tested sample consistency due to the interaction between milk proteins and carrageenan [2].

AFTER

Carrageenan can interact with proteins through electrostatic or ionic bonding. Its molecules carry negative charges and can therefore be combined with positively charged proteins or particles like potassium (c.a. 520 mg), which is found in WPC80. For example, when carrageenan combines with a milk protein such as casein it can form a three-dimensional gel network [43]. Additionally, Gustaw et al. [2] suggested that carrageenan affects the increase in the hardness of dairy desserts, but at a higher concentration it may cause adverse changes in the tested sample texture due to the interaction between milk proteins and carrageenan [2].

We also have added an extended information in lines 393-397 to better explain obtained results:

“This can occur between β-lactoglobulins and ĸ-carrageenan based on depletion interactions in a suspension of small proteins immersed in a solution of polysaccharides that then form an entangled network [26]. These theories can explain the decrease in the back-extrusion measurement value while the hardness of the product increases with the addition of 13% WPC.”

L426 – 430 why is this so?

Thank you for your comment. We have changed this fragment of the manuscript and provided more detailed explanation, hoping that the issue is clarified enough (lines 464-472).

BEFORE

Different antioxidant methods differ in chemical mechanism and may give sometimes different results. However, in the present research the data on DPPH and FRAP method correlated well (correlation coefficient, R=0.84). Some discrepancies between the data from FRAP and DPPH antioxidant methods are not uncommon and occurred before with different biological material [53-55].

AFTER

In the present research, the data obtained from the DPPH and FRAP methods correlated well (correlation coefficient, R=0.84), which is in agreement with many previous studies [53–55]. Some discrepancies between the data from the FRAP and DPPH antioxidant methods occurred. However, such discrepancies are not uncommon and have been reported previously with different biological material [55–57]. Moreover, there are reports in which there is no correlation, or the correlation is weak [58]. Although both the FRAP and DPPH assays are based on the transferring of electrons from the antioxidant and reduction an oxidant, the FRAP method is more prone to color interference and slow color development. It also involves more antioxidant compounds that react with various kinetics [59, 60].

  1. Radzki, W.; SÅ‚awinska, A.; JabÅ‚onska-RyÅ›, E.; Michalak-Majewska, M. Effect of blanching and cooking on antioxidant capacity of cultivated edible mushrooms: A comparative study. Int. Food Res. J. 2016, 23, 599–605.
  2. Wanna, C. Free radical scavenging capacity and total phenolic contents in peel and fleshy crude extracts of selected vegetables. Pharmacogn. J. 2019, 11, 1351–1358, doi:10.5530/pj.2019.11.209.
  3. Thaipong, K.; Boonprakob, U.; Crosby, K.; Cisneros-Zevallos, L.; Hawkins Byrne, D. Comparison of ABTS, DPPH, FRAP, and ORAC assays for estimating antioxidant activity from guava fruit extracts. J. Food Compos. Anal. 2006, 19, 669–675, doi:10.1016/j.jfca.2006.01.003.
  4. Radzki, W.; SÅ‚awiÅ„ska, A.; JabÅ‚oÅ„ska-RyÅ›, E.; Gustaw, W. Antioxidant capacity and polyphenolic content of dried wild edible mushrooms from Poland. Int. J. Med. Mushrooms 2014, 16, 65–75, doi:10.1615/IntJMedMushr.v16.i1.60.
  5. Djordjevic, T.M.; Šiler-Marinkovic, S.S.; Dimitrijevic-Brankovic, S.I. Antioxidant activity and total phenolic content in some cereals and legumes. Int. J. Food Prop. 2011, 14, 175–184, doi:10.1080/10942910903160364.
  6. Gil, M.I.; Tomás-Barberán, F.A.; Hess-Pierce, B.; Kader, A.A. Antioxidant capacities, phenolic compounds, carotenoids, and vitamin C contents of nectarine, peach, and plum cultivars from California. J. Agric. Food Chem. 2002, 50, 4976–4982, doi:10.1021/jf020136b.
  7. Clarke, G.; Ting, K.N.; Wiart, C.; Fry, J. High correlation of 2,2-diphenyl-1-picrylhydrazyl (DPPH) radical scavenging, ferric reducing activity potential and total phenolics content indicates redundancy in use of all three assays to screen for antioxidant activity of extracts of plants from the malaysian rainforest. Antioxidants 2013, 2, 1–10, doi:10.3390/antiox2010001.
  8. Huang, D.; Boxin, O.U.; Prior, R.L. The chemistry behind antioxidant capacity assays. J. Agric. Food Chem. 2005, 53, 1841–1856, doi:10.1021/jf030723c.

L437 – 454 the radar graph is correctly plotted. However, the results needs to be revised to give a clear interpretation of the same and reflect on the data provided in the radar graph. Additional comments on how L437 - 454 needs to be revised can be seen below. L438 – 439 the information needs to be elaborated on to make it clear what you mean

Thank you for your comment. In lines 480-482 (graph description) we have added an information for readers to better understand obtained results:

"Points from 1 to 10 describe the acceptability of the products (Flavor: 1 – the lowest score, 10 - the highest score; Body, texture, appearance and color: 1 – the lowest score, 5 - the highest score).”

In lines 486-488 we have elaborated an explanation about interpretation of provided results:

“Each sample was evaluated using a 1–10-point scale for the description of flavor and a 1–5-point scale for the body, texture, appearance and color of the desserts. The highest point scores related to the greater acceptability of a given feature by panelists.”

L443 please expand on how you determined the differences in the consistency? Did you use the power law which is the optimal way that is recognised to record quantitative consistency data?

Thank you for your comment. We have meant the texture and body (we have left only texture in the sentence), and therefore we have decided to remove the word “consistency” to be more strict and not to mix up with, e.g. the consistency index.

L443 please elaborate on what the highest notes mean in this context.

Thank you for your comment. In lines 486-491 and in the description of radial graph we have added an information about points (scores) and their meanings.

AFTER

“Each sample was evaluated using a 1–10-point scale for the description of flavor and a 1–5-point scale for the body, texture, appearance and color of the desserts. The highest point scores related to the greater acceptability of a given feature by panelists. Panelists evaluated the products objectively. Then the scores received from all study participants for each described feature were added up and averaged.”

L445 please specify which product you are referring to.

Thank you for your comment. We have added an information in line 497:

“…that the product with 9% WPC80 had …”

ADDITIONALLY

We have added an information and explanation of our result to improved “Organoleptic Evaluation” section.

In lines 474-476 we have added an information introducing readers to sensory analysis”

“The sensory characteristics of these high-protein fat-free dairy desserts result from the composition of our product formula. The source of protein and lack of typical fat source had a great impact on the taste and body features of the tested samples.”

In lines 488-491 we have described way we obtained the results shown in the sensory graph:

“Panelists evaluated the products objectively. Then the scores received from all study participants for each described feature were added up and averaged. The dairy dessert …”

In lines 498-500 we have explained obtained results:

“The presence of air bubbles and phase separation in the samples with 13% WPC80 was noted and, because of that, this sample received the lowest scores (Fig. 6).”

In lines 508-517 we have discussed and summarized our results:

“We can confirm these observations. Furthermore, our research provides new information on the texture and taste of dairy desserts. The correlation between the amount of added whey protein and the acceptability of the product was noted.

Based on the sensory evaluation, we can conclude that the intensive taste and smell of WPC80 in our product attracted the most attention from people who tested the fat-free dairy desserts. The liquid texture and high viscosity were also not well received. The homogeneous and uniform consistency of the product with 9% WPC80 represents the product that is the closest to the commercial dairy desserts that the typical consumer is used to eating. We conclude that this made this product the highest-rated sample.”

This previous comment needs to be addressed in the manuscript. L110 – 116 the authors need to account for the effect of the container on the measurements given that carrying out penetration test inside a plastic container there will definitely be an impact from the walls of the container on the measurements. I appreciate the fact that the authors mention that “We have taken into consideration the effect of plastic container, therefore the test was a modified TPA/penetration test (applied to different semi-solid dairy products e.g. processed cheese, cheese sauces, desserts etc.)” but this information needs to be included in the manuscript.

Thank you for your comment. We have added this information to the Results and discussion section (lines 219-223).

“We have taken into consideration that carrying out a penetration test inside a plastic container has an additional impact on the measurements because of the effect of the container walls. However, the penetration test or modified TPA applies to different semi-solid dairy products, e.g., desserts, processed cheese, cheese sauces etc.”

ADDITIONALLY

We have improved “Results and discussion” section:

In lines 217-219:

“Table 2 presents the effect of different WPC80 quantities (5–13%) on the texture attributes (hardness, adhesiveness, cohesiveness and springiness) of high-protein fat-free dairy desserts.”

In lines 258-262:

“They also noticed that in a product with added coconut oil, not only were the values of hardness lower in samples with the highest amount of added whey-protein concentrate compared to a product with a lower amount of WPC80, but also the adhesiveness of the samples. Adhesiveness is a force that is required to remove food from the palate or teeth while chewing [23].”

Conclusions

We have improved “Conclusions” section:

In lines 521-524:

“Generally, the increase of WPC80 content increased the hardness (5–11% WPC80 content), adhesiveness (5–13% WPC80 content), cohesiveness (5–13% WPC80 content), springiness (5–11% WPC80 content) and viscosity (5–13% WPC80 content) of the high-protein fat-free dairy desserts tested.”

In lines 527-530 we have added proposed explanation of the obtained results:

“This may be the result of interactions between the individual ingredients of the product, for example, carrageenan with proteins through electrostatic or ionic bonding, or by phase separation caused by β-lactoglobulins and ĸ-carrageenan interactions.”

In lines 534-536 we have added an information to better conclude organoleptic results:

“At the same time, the use of high quantities of whey proteins makes these proteins very noticeable in the product in terms of taste and smell.”

Thank you very much.

Round 4

Reviewer 1 Report

The revised manuscript reads much better and the authors have taken on board the comments and suggestions which have helped in improving the quality of the paper immensely.